# The kinetics of nsp7-11 polyprotein processing and impact on complexation with nsp16 among human coronaviruses

Kira Schamoni-Kast [1,2,3], Boris Krichel[1,2,3], Tomislav Damjanović[1,2], Fatema-Aqila Said[1], Thomas Kierspel [1], Sibel Toker[1] & Charlotte Uetrecht [1,2] ✉

In coronavirus (CoV) infection, polyproteins (pp1a/pp1ab) are processed into non-structural proteins (nsps), which largely form the replication/transcription complex (RTC). The polyprotein processing and complex formation is critical and offers potential therapeutic targets. However, the interplay of polyprotein processing and RTC-assembly remains poorly understood. Here, we study two key aspects: The order of polyprotein processing by viral main protease M[pro] and its influence on complex formation with the methyltransferase nsp16. Moreover, we establish an approach to determine rate constants $k$ from cleavage sites in structured CoV polyprotein based on native mass spectrometry (MS). The high sensitivity and precision of our method allow quantification of multi-reaction kinetics of nsp7-11 processing from four human pathogenic CoV species. The experimentally determined rate constants are put into perspective with a comprehensive analysis of primary sequences and structural models, revealing distinct cleavage mechanisms for each site based on their local structural environments. Our systematic approach provides a blueprint for kinetic analysis of complex multi-cleavage reactions.

Human coronaviruses (hCoVs) encompass a spectrum of pathogens, from the seasonal alphacoronavirus HCoV-229E to the more virulent betacoronaviruses SARS-CoV, SARS-CoV-2, and MERS-CoV[1]. Two-thirds of the ~ 30 kb single-stranded (+)-sense RNA genome of CoVs encodes proteins primarily forming the viral replication-transcription complex (RTC) and is translated directly into two overlapping polyproteins (pp1a and pp1ab)[2–5]. The polyprotein internal proteases PL[pro] (nsp3) and M[pro] (nsp5) facilitate proteolytic processing of pp1a and pp1ab into mature non-structural proteins (nsps) nsp1-nsp11 and nsp1-nsp16, respectively[6–11]. In particular, M[pro] cleaves nsp5 to nsp16, which are directly involved in viral replication and transcription. While nsp12 and nsp13 harbor the function of the polymerase-helicase complex, the other core enzymes nsp14, nsp15, and nsp16 have an indispensable function in RNA modification. The proteins nsp14 and nsp16 complex with their cofactor nsp10 and function as essential methyltransferases in the viral life cycle[12]. In higher eukaryotes, methylation of the 2'-OH position on the first ribose of RNA transcripts (RNA capping) enables the immune system to distinguish between self and non-self RNA[13,14]. Through its 2'O-methyltransferase activity, the nsp16/10 complex allows viral RNA to evade host immune detection[15].

The polyprotein region nsp7-11 is located at the C-terminal end of pp1a and contains the four small soluble domains nsp7 to nsp10, which have important regulatory roles in replication. It ends with nsp11, a thirteen amino acid peptide lacking known function but being identical in sequence to the overlapping nsp12 N-terminus. Notably, the genetic region encoding nsp11 regulates the ratio of pp1a and pp1ab via genomic frameshift[16–18]. Importantly, the frameshift efficiency makes the translation of pp1a two to five times more likely than pp1ab,

[1]CSSB Centre for Structural Systems Biology, Deutsches Elektronen Synchroton DESY, Leibniz Institute of Virology, University of Lübeck, Hamburg, Germany. [2]Institute of Chemistry and Metabolomics, University of Lübeck, Lübeck, Germany. [3]These authors contributed equally: Kira Schamoni-Kast, Boris Krichel. ✉ e-mail: charlotte.uetrecht@cssb-hamburg.de

thereby controlling the ratio between nsp1-11 (pp1a) and nsp1-16 (pp1ab). The M$^{pro}$-mediated cleavage kinetics control the pathway from complete polyprotein into matured nsps, generating heterogeneous polyprotein or nsp-precursor intermediates. The coordinated processing of the nsp7-10 region is crucial for viral growth[18,19]. The expression of polyproteins and their coordinated processing mechanism has been shown to be an effective viral strategy; however, their detailed processing mechanism in CoVs remains to be elucidated[20–23].

The study of polyprotein processing in both SARS-CoV and SARS-CoV-2 has revealed critical insights into how M$^{pro}$ interacts with and cleaves different sites within the nsp7-11 region. Initial in vitro investigations with SARS-CoV nsp7-10 identified several intermediate processing states, including a relatively stable nsp7-8 product and a transient nsp7-9 intermediate, which helped establish a clear cleavage order, in which cleavage site (CS) CS9/10 was preferentially processed, followed by CS8/9, and finally a significantly slower cleavage of CS7/8[24]. Notably, while M$^{pro}$ could not cleave an isolated CS8/9 synthetic peptide, it efficiently processed this site in the polyprotein context, demonstrating that M$^{pro}$ activity is controlled by both sequence specificity and the broader structural environment. These findings were later corroborated and expanded through integrative structural modeling by Yadav et al., who characterized SARS-CoV-2 nsp7-11 as a flexible ensemble. Referring to the previous work from our group, they confirmed similar processing patterns of CS7/8, CS8/9 and CS9/10 between SARS-CoV and SARS-CoV-2, which was expected given the high sequence conservation between these hCoVs[25]. Further structural evidence from cryo-EM studies by Narwal et al. provided direct visualization of M$^{pro}$ binding to CS9/10 within the nsp7-11 polyprotein. Based on this observation, they suggested that the polyprotein structure could influence M$^{pro}$ targeting to this specific cleavage site[26].

The sequential organization of nsp domains within the polyprotein nsp7-11 region is conserved across known hCoVs[4]. Several studies have examined the M$^{pro}$ cleavage efficiencies, primarily focusing on either screening for ideal protease substrates or measuring activity against synthetic peptides representing polyprotein cleavage sites[27–30]. Hegyi and Ziebuhr investigated the conservation of specificities among CoV proteases by studying four peptides representing CS4/5, CS5/6, CS8/9, and CS9/10. Their findings revealed that M$^{pro}$ from *Alphacoronaviruses*, HCoV-229E and transmissible gastroenteritis virus (TGEV), share similar specificities[27]. However, mouse hepatitis virus (MHV), a distinct *Betacoronavirus*, exhibited different specificities. Our group compared nsp7-8 processing and nsp7 + 8 protein complex stoichiometry and topology in seven CoV species[31]. Notably, distinct assemblies have been observed among different CoVs. However, the conservation of processing mechanisms across CoV species remains poorly characterized.

Understanding the molecular mechanisms of the CoV processing reactions is crucial for antiviral drug development, as it enables the identification of rate-limiting steps and critical interaction sites that could be targeted by therapeutic interventions[32]. To achieve this goal, we need methods that can provide comprehensive information on polyprotein cleavage products and the kinetics of this multi-cleavage processing reaction.

Here, we investigate the kinetics of in vitro processing of nsp7-11 using native mass spectrometry (MS). Native MS is a highly sensitive method allowing simultaneous monitoring of cleavage products, making it very versatile to study dynamic processes and heterogeneous samples[33,34]. In addition, native MS preserves protein-protein interactions, enabling detection of protein complexes formed by the processing products[35]. By acquiring mass spectra at different time points, we monitored increasing and decreasing nsps, intermediate products and protein complexes simultaneously. Based on the change in signal ratio, we established an approach to extract kinetic rate constants $k$ in a multi-cleavage reaction. Thereby, we determined $k$ of M$^{pro}$ at CS7/8, CS8/9, CS9/10 and CS10/11 for human

pathogenic SARS-CoV-2, SARS-CoV, HCoV-229E and MERS-CoV. In additional experiments, we studied processed and unprocessed polyprotein interaction with the SARS-CoV-2 methyltransferase nsp16, identifying critical and non-critical cleavage sites and a chimeric complex between SARS-CoV-2 and MERS-CoV proteins. Analysis of primary sequences, structural models, and experimentally determined rate constants revealed molecular mechanisms of CoV processing.

## Results

To investigate SARS-CoV-2 polyprotein processing, we expressed recombinant nsp7-11 polyprotein and cleaved it with SARS-CoV-2 M$^{pro}$. Initial experiments used substrates with N- or C-terminal His$_6$-tags (nsp7-11N and nsp7-11C, respectively). However, the potential effects of these tags on M$^{pro}$ cleavage activity could not be excluded. Therefore, we also employed tag-free nsp7-11 substrates using an N-terminal His$_6$-SUMO-tag strategy to compare the four hCoVs. This tag was removed using SUMO-specific protease ULP-1, generating nsp7-11 with native termini (Fig. 1A).

We used native MS to analyze SARS-CoV-2 M$^{pro}$ cleavage reactions with nsp7-11 substrates from four hCoVs: SARS-CoV-2, SARS-CoV, HCoV-229E and MERS-CoV. Native MS confirmed that all four conserved cleavage sites (CS7/8, CS8/9, CS9/10 and CS10/11) were addressed in each of them. Despite the wide mass range spanning from the smallest product nsp11 (1325.654 Da ± 0.001 Da) to the heterotetrameric complex of nsp7$_2$ + 8$_2$ (62,244 Da ± 2 Da), we obtained precise masses from tagged and untagged substrates for monomeric nsps, intermediates and polyproteins (Figs. 1C and S1, Supplementary Table S1–S5). The processing reaction was additionally verified by SDS-polyacrylamide gel electrophoresis (SDS-PAGE) (Figs. 1B and S2) as done in previous studies[24,25].

For detailed investigation of M$^{pro}$ mediated polyprotein processing, we followed two approaches to extract fast and slow cleavage rates (Fig. 1D). For fast rates, we used a continuous monitoring approach, where the ongoing enzyme reaction was performed "in-capillary" at 27 °C. For slow rates at 0 °C (on ice), we used a discontinuous approach, with processing reactions performed in a test tube and sampled discontinuously over time. Using a Python-based script, we extracted rate constants $k$ of CS7/8, CS8/9, CS9/10 and CS10/11. Finally, we investigated how processing influences CoV core enzyme complex formation with recombinantly produced SARS-CoV-2 nsp16 and the cleaved as well as uncleaved polyprotein nsp7-11 (Fig. 1E).

Native MS enables monitoring of reaction products over time, though capturing sufficient data points from rapid reactions presents a challenge. To overcome this limitation and observe early M$^{pro}$-mediated processing reactions, we implemented continuous "in-capillary" analysis using nsp7-11C and nsp7-11N substrate polyproteins. The capillary housing temperature was 27 °C, which approximates physiologically relevant viral propagation temperatures. This approach also ensures identical reaction start points, which is essential for extracting high-quality kinetic data[36]. We monitored a decrease of intensity of the polyproteins, nsp7-11C (60,950 Da ± 4 Da) or nsp7-11N (61,085 Da ± 1 Da), and an increase of the cleavage intermediates nsp7-10 (dominant intermediate and plateauing in nsp7-11N after 5-10 min), nsp7-9, nsp7-8, nsp9-11, nsp9-10, and nsp10-11 within the first 30 min of the reaction (Figs. 2A and S3, Supplementary Table S1). This enabled us to monitor the fastest cleavage reaction of CS10/11 in detail (Fig. 2B).

To determine the rate constant $k$ for the CS10/11 cleavage reaction, we developed a custom data processing approach. Signal intensities (area under the curve, AUC) of substrate and intermediate products were plotted over time. Signals of substrate and observed intermediates containing the intact cleavage site (nsp9-11, nsp10-11) were summed and plotted logarithmically. The initial linear slope shows that first order kinetics can be applied and hence $k$ at CS10/11 at 27 °C is calculated from the slope of the linear region (Fig. 2C). Due to

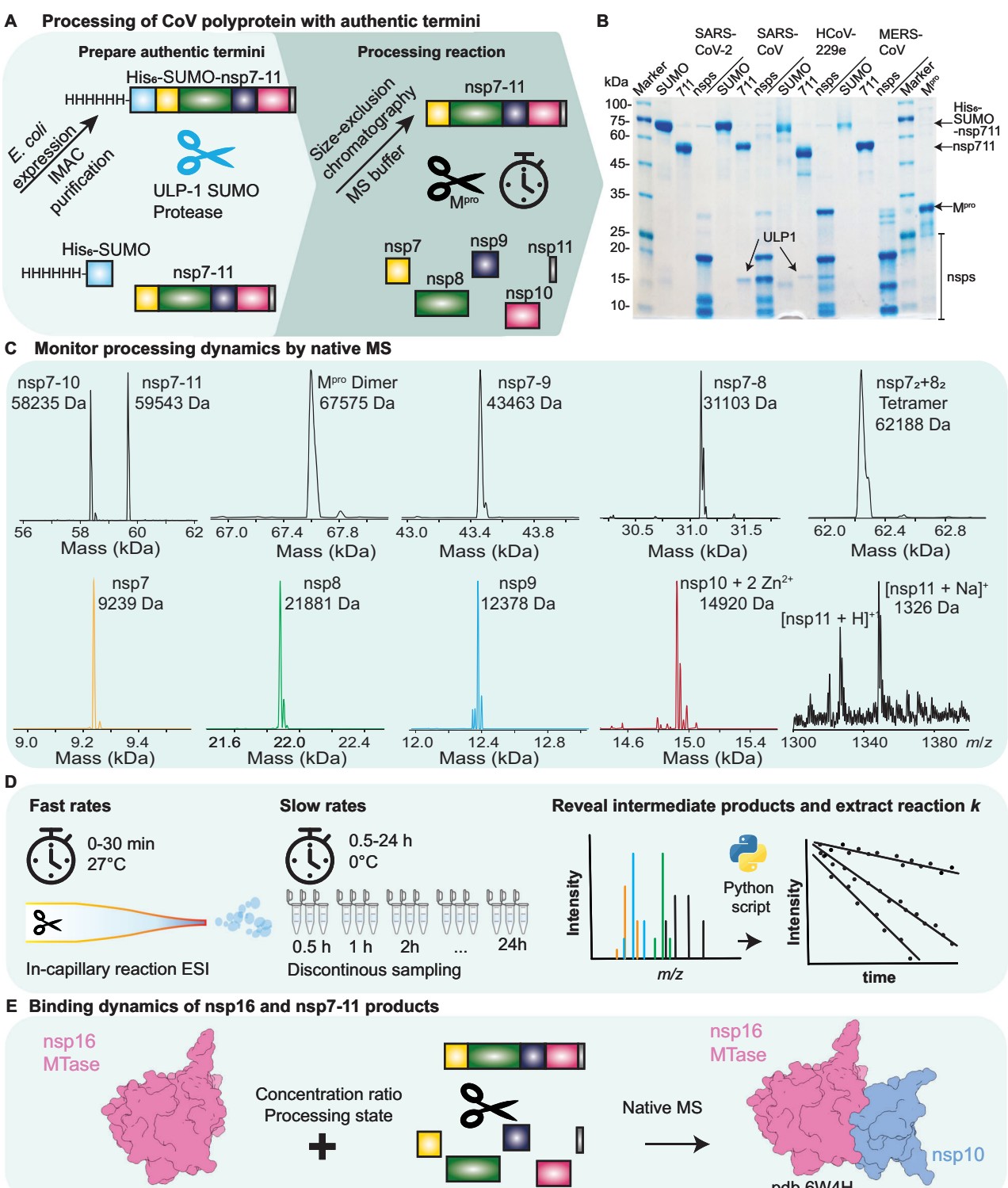

**Fig. 1 | Graphical description of sample preparation and experimental implementation. A** Small ubiquitin-like modifier (SUMO)-His$_6$-tagged protein constructs were recombinantly expressed in *E. coli* and affinity purified. Prior to mass spectrometry (MS) analysis, the samples were exchanged into the volatile MS buffer surrogate ammonium acetate to prevent salt adduct formation. Furthermore, polyprotein processing was conducted by adding recombinantly expressed main protease (Mpro). **B** For comparison with native MS data, samples were collected for sodium dodecyl-sulfate polyacrylamide gel electrophoresis (SDS-PAGE) at the indicated time points. **C** Shows observed species in severe acute respiratory syndrome coronavirus 2 (SARS-CoV-2), including all mature non-structural proteins (nsp). Mature nsps were deconvoluted and are presented with their masses in Dalton (Da). **D** In order to capture the fast and slow rate constants of the multi-cleavage reaction, two measurement approaches were employed, in-capillary reaction (continuous approach) and discontinuous, discrete time-resolved approach, respectively. **E** For probing the interaction between the polyprotein and nsp16, the methyltransferase (MTase), Mpro was added to process the polyprotein (cleaved) or left untreated (intact). An uncropped image of the SDS-PAGE gel can be found in the source data file. Source data are provided as a Source Data file.

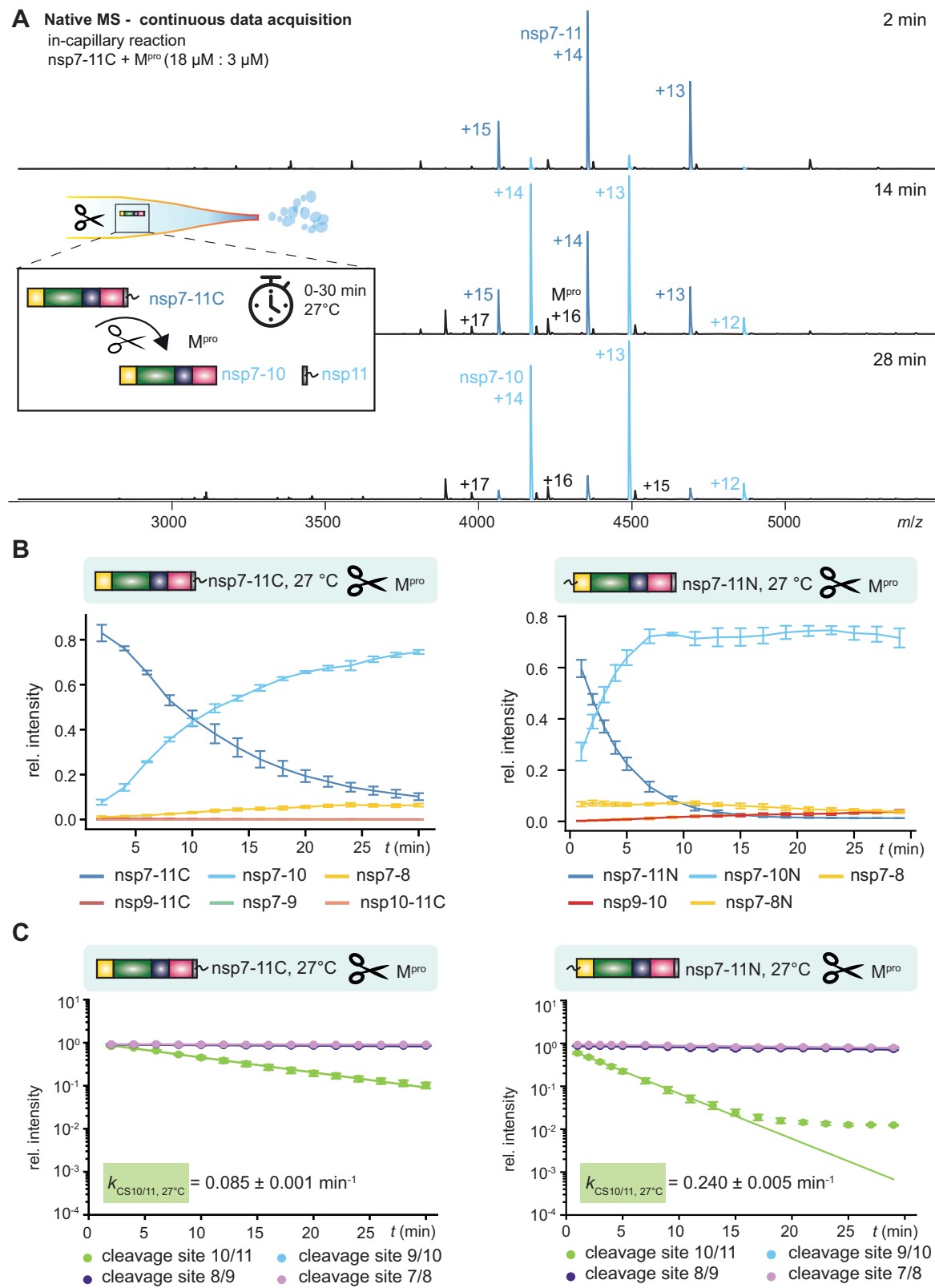

**Fig. 2 | Comparison of continuous processing of severe respiratory syndrome coronavirus 2 (SARS-CoV-2) non-structural protein 7-11 C (nsp7 11 C) and nsp7-11N.** 18 μM nsp7-11 with N- or C-terminal His₆-tag were mixed with 3 μM main protease (Mᵖʳᵒ) at 27 °C. Standard errors were calculated from triplicates. **A** Representative mass spectra of nsp7-11C showing continuous in-capillary processing at 2 min, 14 min and 28 min. **B** Decays of intermediate species of

nsp7-11C and nsp7-11N are plotted and dominated by nsp7-11C/nsp7-11N (blue) and nsp7-10/nsp7-10N species (turquoise). Data points were simply connected for enhanced visibility. **C** Decline of cleavage site 10/11 (CS10/11) and CS9/10 for nsp7-11C and nsp7-11N were plotted on a logarithmic scale, and data points were fitted for the linear region using first-order kinetics. Source data are provided as a Source Data file.

the low intensities below 1% of nsp9-11 and nsp10-11, the rate constant is dominated by the decrease of nsp7-11C or nsp7-11N (Figs. 2A and S3). Therefore, in the first 30 min, the main conversion is the cleavage of nsp11 with CS10/11 in both substrates. However, CS10/11 in nsp7-11 N ($k_{CS10/11, 27\,°C}$ = 0.240 min$^{-1}$ ± 0.005 min$^{-1}$) was cleaved three times faster than in nsp7-11 C ($k_{CS10/11, 27\,°C}$ = 0.081 min$^{-1}$ ± 0.001 min$^{-1}$), indicating that the His$_6$-tag decreases cleavage site efficiency in the latter. Linearity of the slope for CS10/11 breaks down when signal intensities decreased to 10% of their initial value, which occurred after 15 min for nsp7-11N and ~30 min for nsp7-11C. This deviation from linearity likely resulted from increased substrate depletion at 27 °C, whereas at 0 °C, similar substrate depletion would only be expected after longer incubation times of several hours.

The rate constants for the other cleavage sites (CS9/10, CS8/9, CS7/8) could not be determined accordingly because their corresponding products remained of low staggering intensity (14-48-fold slower than CS10/11). However, nsp11 or the His$_6$-tag do not influence either the order of processing or the conversion rates of the other CS (Supplementary Fig. S4). While this continuous processing approach effectively captures early reaction kinetics, it faces limitations for extended monitoring most likely due to limited protein stability at 27 °C and acidification within the capillary over time[37,38]. Determining $k$ for the other cleavage sites required experiments spanning longer time periods, which could only be conducted at a lower temperature. Therefore, we followed up with the discontinuous approach.

A more complete picture emerges using such a discontinuous approach. To further avoid any influence of the His$_6$-tag, untagged nsp7-11 with native termini was employed. We monitored M$^{pro}$-mediated processing of nsp7-11 substrates from four hCoVs. Reactions were conducted on ice to ensure protein stability, and native mass spectra were acquired in triplicate at discrete time points between 30 min and 500 min. For SARS-CoV-2 nsp7-11 processing, we monitored a dynamic succession of cleavage products. Expectedly, the intensity of the cleavage intermediate nsp7-10 peaks first. It quickly gives rise to subsequent cleavage intermediates nsp7-8, nsp7-9, and nsp9-10. At later time points, when the substrate is depleted also lower populated species like nsp9-11 become visible (Fig. 3A). We validated the reaction after 24 h, showing mainly monomeric nsps, indicative of nearly complete cleavage (Supplementary Fig. S5). Both tagged and untagged substrates produced corresponding products and revealed similar cleavage order (Supplementary Fig. S4). However, all kinetic parameters were determined exclusively from experiments using untagged substrates to avoid any potential tag-induced artifacts.

To extract kinetic rate constants, we again applied our custom data processing to simplify the multiple cleavage reactions to first-order kinetics. Peak areas of intermediate species were summed based on their intact cleavage sites (Fig. 3B). These summed values were fitted with an exponential decay (Fig. 3C). Plotting the y-axis on a logarithmic scale allows verification of first-order kinetics through linear fits (Fig. 4A). Data of the other three hCoVs (Supplementary Figs. S5–S8) was processed (Supplementary Fig. S9) and fitted accordingly (Fig. 4).

During the experiment, we checked with collision-induced dissociation whether peaks originated from the intermediate product nsp7-8 or nsp7 + 8 heterodimer. The SARS-CoV-2 precursor ion $m/z$ 3110 did not dissociate into nsp7 and nsp8, meaning <1% product ion signal intensity compared to precursor ion intensity originates from the heterodimer. Thus, nsp7 and nsp8 are still too low abundant for complexation and only become significantly populated between 6 h and 24 h.

In SARS-CoV, the dominant products next to nsp7-10 were nsp7-9 and nsp7-8 early in the reaction, nsp9-11 also becomes visible later on (Supplementary Figs. S6 and S9). The observed relative intensities of intermediate cleavage products are largely similar to SARS-CoV-2 nsp7-11, suggesting a similar processing pattern from C- to N-terminus, consistent with previous studies[24]. This similarity is reflected in the

comparable order of rate constants, although SARS-CoV showed slower cleavage at CS7/8 and faster cleavage at CS9/10 compared to SARS-CoV-2 (Fig. 4). In HCoV-229E, the observed dominant early intermediate was nsp7-9. While nsp7-8 and nsp10-11 are observed at later time points, they never make up a relevant share of the intensity (Supplementary Figs. S7 and S9). While some early observed products from HCoV-229E nsp7-11 resemble those observed in SARS-CoV and -2, nsp7-10 is essentially absent. It is tempting to state that CS10/11 is hence not addressed first. However, the lack of populated nsp10-11 in the early phases of the reaction rather suggests that CS9/10 and CS10/11 are processed at similar rates, which is corroborated by the linear fits (Fig. 4). MERS-CoV exhibited the most distinct intermediate distribution, with nsp7-8 and nsp9-11 emerging as dominant species from the onset throughout the reaction over 500 min (Supplementary Figs. S8 and S9). This unique intermediate pattern effectively results in MERS-CoV nsp7-11 being processed 'in half' at CS8/9. The data suggests early and efficient cleavage at the CS8/9 site, while CS9/10, CS10/11 and CS7/8 all showed similarly retarded cleavage rates, a pattern distinct from the other three hCoVs (Fig. 4). Indeed, the rate constant for CS8/9 cleavage in MERS-CoV ($k_{cs8/9, 0°C}$) is approximately twice as fast as any cleavage site rate constant in the other hCoVs. The other MERS-CoV cleavage sites are not processed slowest among hCoVs tested, as the rate constants of cleavage sites CS7/8 ($k_{cs7/8, 0°C}$) in the other hCoVs are two to thirty times slower.

Native MS revealed distinct processing patterns across the four hCoVs. SARS-CoV and SARS-CoV-2 identified CS10/11 as the dominant early cleavage site, while HCoV-229E and MERS-CoV exhibited different patterns. Despite identical sequences at CS7/8 in both SARS species, their rate constant differed by an order of magnitude, suggesting structural rather than sequence effects on cleavage efficiency. The core residues P2 and P1 (L and Q) are conserved within CS7/8, 8/9 and 9/10 across all species, yet different rates were observed, particularly at CS8/9 between HCoV-229E and MERS-CoV, indicating that flanking sequences or structure influence processing. For CS9/10, where the P4 to P3' positions are identical across all species, the varying cleavage rates likely result from structural differences. CS10/11 showed the greatest sequence variability, especially at P2, with MERS-CoV containing proline and HCoV-229E containing isoleucine, potentially explaining their slower kinetics. These results reveal that C- to N-terminal processing of nsp7-11 is not conserved among SARS-CoV, SARS-CoV-2, HCoV-229E, and MERS-CoV, though delayed CS7/8 cleavage appears to be a common feature. The non-essential nature of fast CS10/11 cleavage raises the question whether uncleaved intermediates can still function as cofactors in complex formation.

Formation of the RTC requires processing, but whether the RTC incorporates exclusively mature nsps or also immature processing intermediates remains unknown. The functional RTC requires association of several proteins, including nsp10 and nsp16[12,19,39]. We hypothesize that nsp16 + 10 complex formation similarly depends on polyprotein processing, specifically the cleavage of CS9/10 and, to a lesser extent, CS10/11 to release nsp10 from the polyprotein. To test this hypothesis, we performed protein-protein interactions using native MS (Fig. 5 and S10). Initially, we tested binding of uncleaved SARS-CoV-2 or MERS-CoV nsp7-11 (59674 Da ± 3 Da and 59658 Da ± 4 Da, respectively) and SARS-CoV-2 nsp16 (33323.27 Da ± 0.14 Da). For SARS-CoV-2 nsp7-11, predominant signal intensities originated from nsp16 monomer, nsp7-11 monomers and dimers, and low intensities for nsp7-11 + nsp16 complex (~2%). Increased levels (<5%) of SARS-CoV-2 nsp16 complexed with MERS-CoV nsp7-11 were observed despite being a chimeric complex. The complexes were validated using collision-induced dissociation (Supplementary Fig. S11), which notably revealed the Zn$^{2+}$ binding of nsp10[40].

We then initiated processing of nsp7-11 by adding M$^{pro}$ and incubating with nsp16 overnight. Native mass spectra were distinct for SARS-CoV-2 and MERS-CoV nsp7-11, although both showed high levels of

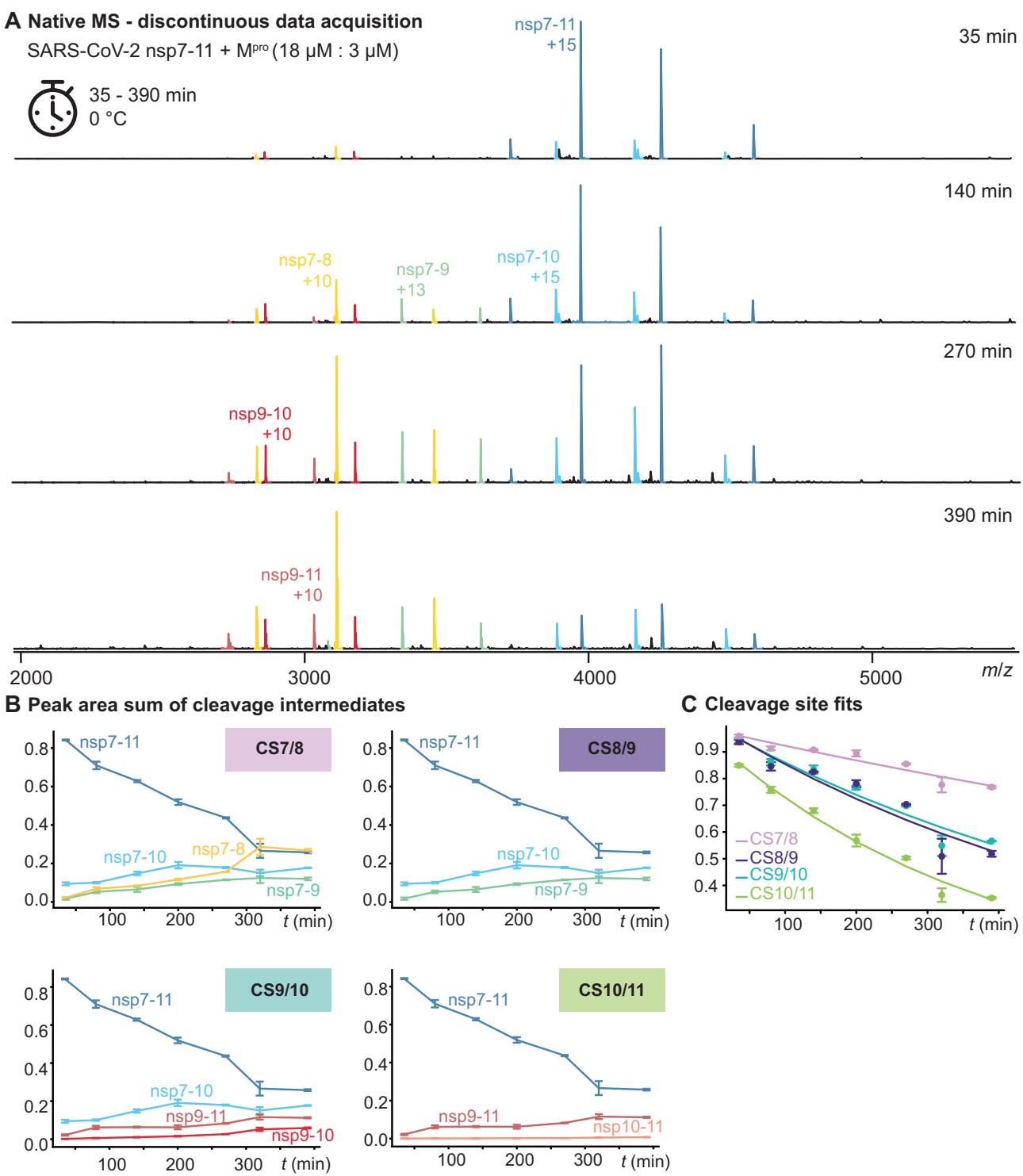

**Fig. 3 | Discrete processing approach of severe respiratory syndrome coronavirus 2 (SARS-CoV-2) non-structural protein 7-11 (nsp7-11).** 19 μM nsp7-11 was mixed with 3.5 μM M^pro at 0 °C. A discrete processing approach was conducted in triplicate measurements. Error bars are standard error. **A** Representative mass spectra at different time points. **B** Course of the individual intermediate species of the polyprotein assigned to the four corresponding cleavage sites (CS) (indicated on the top right). Data points are connected for better visibility. **C** Determination of the rate constants $k$ by following the depletion of the substrates corresponding to each cleavage site. The decay is represented as a fitted line. Source data are provided as a Source Data file.

complexation between SARS-CoV-2 nsp16 and nsp10, suggesting specific binding. A heterodimeric complex containing mature nsp10 (nsp16 + 10, 48,236 Da ± 1 Da) was apparent in SARS-CoV-2 (Supplementary Table S6). Although $k_{CS10/11, \, 0°C}$ in MERS-CoV would suggest complete processing overnight, we observed more than 10% nsp10-11 intermediates and more than 40% heterodimeric nsp16 + 10-11 as

protein complex. Strikingly, no nsp16 with mature nsp10 was observed, indicating that nsp16 binds to nsp7-11 or nsp10-containing intermediates, which potentially protects CS10/11 from further cleavage in the complex. Given the moderate sequence similarity between MERS-CoV and SARS-CoV-2 (70% for nsp10 and 80% for nsp16), the formation of chimeric nsp16 + 10 complexes represents an intriguing finding.

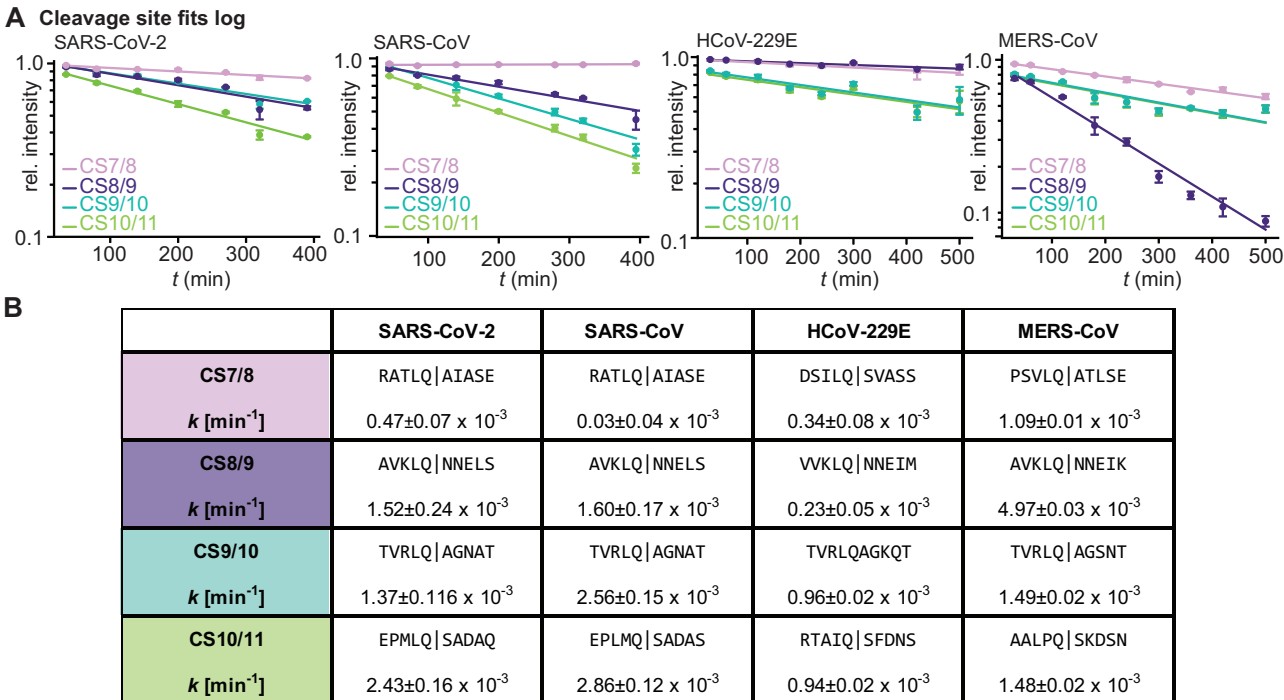

**Fig. 4 | Kinetic rates of all four coronaviruses (CoVs): Severe acute respiratory syndrome CoV 1, SARS-CoV-2, middle east respiratory syndrome CoV (MERS-CoV), and human CoV-229E (HCoV-229E).** Experiments were conducted in triplicates and error bars are standard error. **A** Relative intensities of respective substrate species were summed for each cleavage site (CS) and plotted over time on a logarithmic scale, resulting in a linear fit illustrating first-order kinetics of the otherwise exponential fitting model. The standard error of the mean (SEM) is also provided. **B** Extracted kinetic rates for each CS $k$ in min$^{-1}$ at 0 °C. Source data are provided as a Source Data file.

To mimic the viral ratio of pp1a to pp1ab, we tested increased proportions of cleaved SARS-CoV-2 nsp7-11 to nsp16, observing similarly increased proportions of nsp16 + 10 complex formation (Supplementary Fig. S12). These experiments yielded a complex dissociation constant $K_D$ of 8 μM ± 1 μM. In comparison, titration measurements of purified recombinant nsp10 and nsp16 yielded a lower $K_D$ of 1.4 μM[40]. The higher $K_D$ value observed here may result from the complex mixture of polyprotein cleavage products in our experimental system, which could lead to signal suppression for the complex[41,42]. In summary, nsp16 showed weak binding to immature nsp10 within the polyprotein but strong binding to mostly mature nsp10. Our results indicate complex formation requires N-terminal CS9/10 cleavage but not necessarily C-terminal CS10/11 cleavage. Available crystal structures of nsp16/10 cannot explain this cleavage site preference, as both nsp10 termini are distant from the nsp16 binding site (Fig. 5C). We conclude that while complete processing of nsp7-11 is not essential, it greatly enhances CoV methyltransferase nsp16 + 10 complex formation.

## Discussion

In this work, we characterized the nsp7-11 polyprotein processing kinetics of four hCoV species SARS-CoV, SARS-CoV-2, HCoV-229E and MERS-CoV. Using native MS, we quantified multi-reaction kinetics and determined rate constants $k$ for all four contained M$^{pro}$ cleavage sites simultaneously. Our analysis revealed both conserved and unique features in nsp7-11 processing reactions. In addition, we demonstrated that while complete processing of nsp7-11 is not essential, it greatly enhances methyltransferase complex assembly. Here, we discuss structural implications of our findings and evaluate our approach against conventional techniques.

To investigate how protein sequence and structure relate to the conversion rates of the four substrates, we analyzed structural models generated by AlphaFold3[43-46] (Fig. 6 and Supplementary Figs. S13 and S14). AlphaFold3 indicates values of the predicted local distance difference test (pLDDT), which measures the confidence per residue of local structure prediction, estimating how well the prediction would agree with an experimental structure. These local confidence scores are stored in the B-factor column of output files, allowing their visualization. The predicted nsp7-11 models across all four hCoVs showed that the nsp domains nsp7, nsp8, nsp9, and nsp10 were predicted as folded proteins and had similar local confidence scores usually above 70, indicating medium to high confidence. The pLDDT-values at the cleavage sites and nsp11 regions were below 60, indicating rather low confidence (Supplementary Figs. S15–18) and accordingly were predominantly predicted as disordered regions. An exception was the CS7/8 cleavage site region, which was predicted as an α-helical structure across all species. Here, the pLDDT-value fluctuates between 30-50, representing a low confidence. Thus, the α-helix is not a confident prediction, but together with our experimental data, provides a logical explanation for the slow kinetic rates of CS7/8. Next to the pLDDT value, the predicted aligned errors (PAE) are estimating the confidence of how well two residues of two different domains are placed within the predicted structure. PAE-values show low confidence considering the spatial organization of the individual domains, pointing to a flexible spatial organization of nsp7-11 domains (Supplementary Fig. S19). Altogether, the nsp7-11 models resembled beads on a string, with globular nsp domains linked by flexible cleavage sites, consistent with structures suggested from integrative modeling and previous SARS-CoV cleavage results[24,25].

Analysis of the determined conversion rates revealed notable correlations with structural predictions. The largest variation in conversion rates occurred at CS8/9, CS9/10 and CS10/11, where Alpha-Fold3 predicted disorder in the corresponding linker regions. Across all four strains, CS7/8 had relatively slow cleavage rates and was predicted α-helical (Fig. 6). As α-helices generally serve as poor protease substrates, a significant structural transition would be required

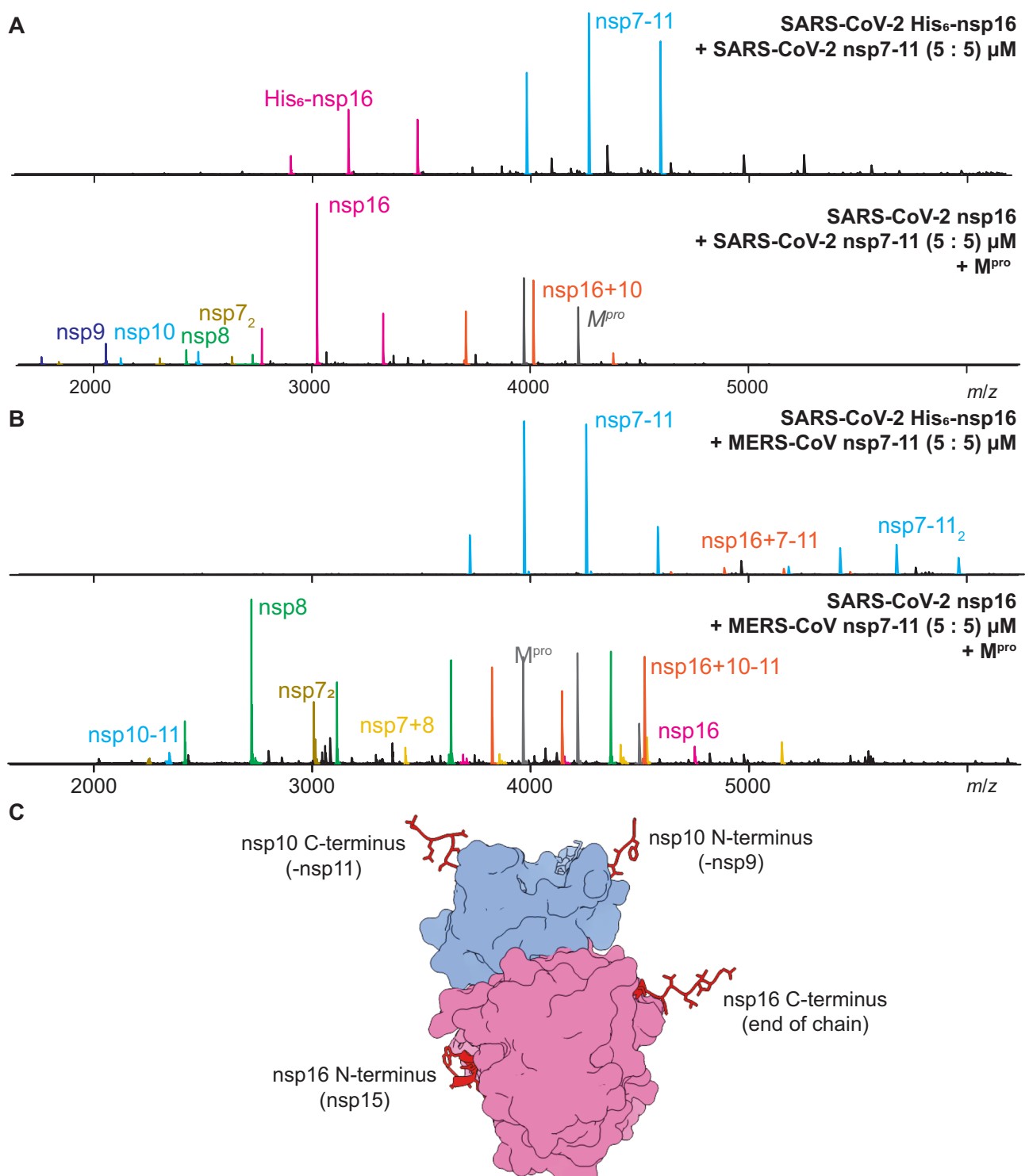

**Fig. 5 | Non-structural protein 16 (nsp16) protein-protein interaction with nsp7-11 polyprotein. A** The first mass spectrum shows severe acute respiratory syndrome coronavirus 2 (SARS-CoV-2) nsp16-His$_6$ (pink) mixed with SARS-CoV-2 nsp7-11 (blue) and low intensity binding of nsp16 to nsp7-11. Intensity of nsp16 to nsp7-11 is so low that it is not visible in the depicted spectrum. The second spectrum shows a mixture of nsp7-11, nsp16-His$_6$ and the main protease (M$^{pro}$). M$^{pro}$ had fully processed nsp7-11 and thereby liberated nsp10 (blue), which is a cofactor of nsp16 (pink). The binding interaction of nsp16 + 10 (orange), mature nsps nsp7$_2$ (olive), nsp8 (green), nsp9 (dark blue) and nsp10 (blue) are shown. **B** Top mass spectrum shows Middle East respiratory syndrome CoV (MERS-CoV)

nsp7-11 (blue) mixed with SARS-CoV-2 nsp16-His$_6$. A low-abundant nsp7-11+nsp16-His$_6$ complex (orange) could be detected. The bottom mass spectrum shows SARS-CoV-2 nsp16 (pink) mixed with fully processed MERS-CoV nsp7-11 by M$^{pro}$, resulting in released nsp7$_2$ (olive), nsp8 (green), nsp9 (dark blue) and nsp10-11 (blue). Nsp16 + nsp10-11 complex (orange) and nsp7 + 8 (yellow) are detected. **C** Surface rendered crystal structure (pdb 6W4H) of nsp16 (pink) and nsp10 (blue) illustrates the spatial orientation of C- and N-termini of nsp10 and nsp16 pointing away from each other. Binding experiments for both SARS-CoV-2 and MERS-CoV nsp7-11 were repeated twice. Source data are provided as a Source Data file.

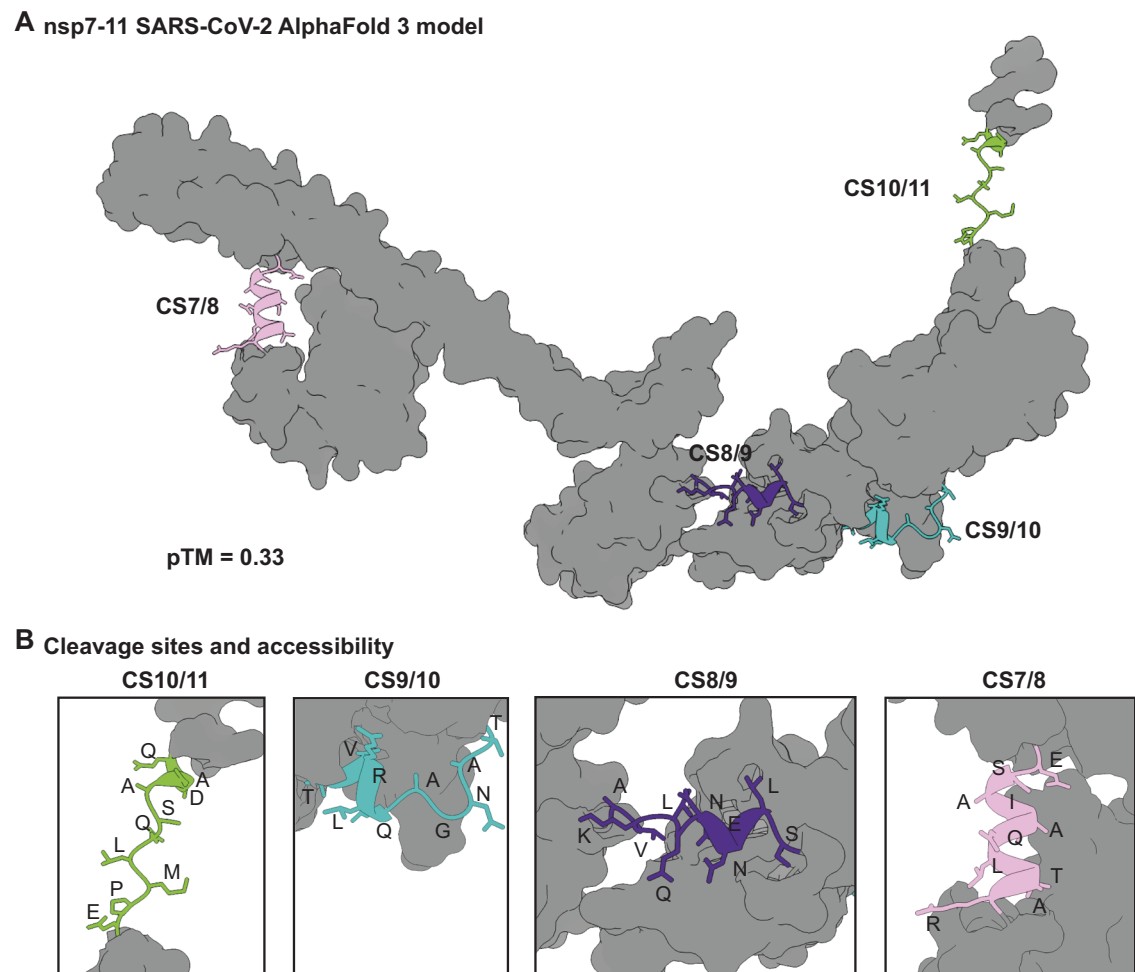

**Fig. 6 | Structure prediction by Alphafold3 of SARS-CoV-2 nsp7-11 polyprotein.** **A** Predicted overall folding of nsp7-11 of SARS-CoV-2 (model 0) with four cleavage sites (CS): CS7/8 (pink), CS8/9 (purple), CS9/10 (cyan) and CS10/11 (green). **B** Zoom in on the four CS with labeled amino acids illustrating predicted accessibility. Source data are provided as a Source Data file.

explaining slower kinetics[47–49]. This is further supported by CS7/8 adopting a disordered conformation when crystallized with M$^{pro}$, indicating that the structure indeed has to adapt to the protease binding grove[50,51]. These findings help explain the observed differences in conversion rates: The disordered regions at CS8/9, CS9/10, and CS10/11 can rapidly undergo structural changes to accommodate M$^{pro}$, while the stable α-helical structure at CS7/8 requires more reorganization and more time to fit the protease active site. Our $k_{CS7/8,\ 0°C}$ suggest however, that this α-helical structure should exhibit distinct unfolding dynamics with lower stability in MERS-CoV and higher stability in SARS-CoV. Moreover, nsp8 undergoes partial conformational changes during processing, which may contribute to the observed delayed cleavage kinetics at its associated cleavage sites[25].

A closer examination of primary sequences, predicted structures, and our determined conversion rates provided insights into the cleavage mechanism at each site. Across all tested species, cleavage sites CS7/8, CS8/9, and CS9/10 contained the typical M$^{pro}$ consensus sequence elements: L at P2 and Q at P1[28,52–54]. Despite these primary sequence similarities, their conversion rates varied significantly. For CS7/8, as discussed above, its secondary structure appears to be the key factor inhibiting cleavage. At CS9/10, all tested strains shared sequence conservation from P5 to P2', and cleavage occurred with relatively high efficiency. Previous studies demonstrated that residues up to P6 in CS9/10 interact tightly with M$^{pro}$, explaining the consistent processing efficiency across strains at this site. The conversion rates of CS8/9 showed unexpected variability between strains, exhibiting the

highest measured rate in MERS-CoV but slow rates in HCoV-229E. This variability was particularly surprising given its highly conserved non-canonical sequence, notably the NNE at P1'-P3' essential for nsp9 post-translational modifications in CoV transcription[55]. The only differences among species at CS8/9 specific to HCoV-229E that could explain an altered interaction with M$^{pro}$ was a P5 A-to-V substitution, suggesting these substantially inhibit CS8/9 cleavage. MERS-CoV, however, shares the P5-P1 with SARS-CoV and -2, suggesting that other flanking residues or structural aspects determine this highest conversion rate. Hence, our findings indicate that variations in conversion rates likely arise from specific structural features within or surrounding the cleavage sites, rather than primary sequence alone.

It is important to note that all nsp7-11 substrates from the four hCoVs (SARS-CoV-2, SARS-CoV, HCoV-229E and MERS-CoV) were cleaved using SARS-CoV-2 M$^{pro}$. While M$^{pro}$ consensus sequences are generally conserved across coronaviruses, species-specific differences in protease-substrate interactions cannot be entirely excluded[27,53,56]. Future studies using matched species-specific M$^{pro}$-substrate pairs could further refine our understanding of these processing kinetics. However, this would further complicate assigning observed differences to specific features in the polyproteins.

The CS10/11 provided particularly valuable insights. CS10/11 showed high-efficiency cleavage in SARS-CoV and SARS-CoV-2, but low efficiency in HCoV-229E and in MERS-CoV. Importantly, this low efficiency CS10/11 proved non-essential for the formation of the RTC sub-complex nsp16 + 10-11. CS10/11 showed relatively low sequence

conservation among tested species, with notable substitutions at the P2 position of the M[pro] consensus sequence. In this position, the typically conserved L is replaced with M in SARS-CoV, I in HCoV-229E, and P in MERS-CoV. A detailed study of M[pro] substrate specificity supports our findings, showing that M at position P2 still permits moderate cleavage efficiency, while other substitutions result in lower efficiency[29]. In our structural models, CS10/11 occupies a peripheral, exposed position, which likely enables M[pro] substrate recognition despite unfavorable sequence motifs. Interestingly, hydrogen-deuterium exchange experiments revealed that nsp11 shields its adjacent nsp10 region from deuterium uptake, suggesting it effectively covers this interface in the native structure. However, our findings suggest that CS10/11 cleavage efficiency is neither conserved across hCoVs nor required for complexation with nsp16, possibly reflecting CS10/11's origin as a secondary product from the coronavirus ORF1a/ab RNA frameshift region. Notably, in the one-third of coronavirus polyproteins expressed as pp1ab, nsp10 is followed by nsp12. Since nsp12 shares the same N-terminus as nsp11, the cleavage sites CS10/11 and CS10/12 are structurally similar by design. This raises the intriguing possibility that a long-lived nsp10-12 intermediate could exist and form a complex with nsp16, resulting in an nsp10-12 + 16 super-complex.

Our research illuminates the sophisticated relationship between CoV polyprotein processing and nsp complex formation. Notably, functional chimeric complexes can even form between components from different CoV species, such as SARS-CoV-2 and MERS-CoV. Spatiotemporally coordinated processing generates multiple proteoforms with potentially distinct functions and is a common pattern in viruses[21,57]. The coronavirus methyltransferase complex demonstrates remarkable flexibility, while it binds unprocessed polyprotein nsp7-11, it clearly prefers mature and half-mature products nsp10 or nsp10-11. The integrative model of Yadav et al. for SARS-CoV-2 nsp7-11 suggests that nsp9 and nsp10 are in close proximity, which could explain why the polyprotein is a poor binder of nsp16[25]. Our observation of consistently delayed CS7/8 cleavage across all species hints at a possible regulatory mechanism for the assembly of coronavirus polymerase complexes. This spatiotemporal coordination likely orchestrates the sequential formation of various functional assemblies: the processivity-enhancing nsp7 + 8 subunits would join nsp12 later to form the polymerase complex, while nsp9 and critical nsp10-dependent complexes, including the proofreading nsp14/10, methyltransferase nsp16/10, and the recently reported ternary complex nsp10 + 14 + 16, would become available earlier in the viral lifecycle[58,59]. Such regulated processing would ensure that RNA capping and proofreading are in place first, potentially allowing the virus to fine-tune RNA synthesis and modification. The ordered and regulated polyprotein processing parallels cleavage processes in *Alphaviruses*, another group of enveloped positive-sense RNA viruses[60]. In all genera, *Alphacoronaviruses*, *Betacoronaviruses* and *Alphaviruses*, timely and precise polyprotein processing is crucial for viral replication, yet depends on factors beyond sequence accessibility[61,62].

A key advantage of our approach was the use of folded polyprotein substrate, presenting cleavage sites in their native structural context. By using tagless nsp7-11 with natural amino acid sequence, we aimed to replicate authentic M[pro]-polyprotein interaction dynamics. In contrast, conventional techniques typically use either natural libraries or artificial substrates such as labeled or unlabeled peptides or short cleavage site sequences expressed between reporter proteins. These methods offer advantages in throughput and automated readout, enabling broad sequence space and condition sampling[29,63,64]. In a previous study, we have shown that peptide-based cleavage sites may not reflect native structural dynamics, potentially leading to misleading results[24]. For example, FRET-based assays of SARS-CoV and MERS-CoV suggested high conversion rates for CS7/8, indicating a different processing order than in our findings[20,30]. We therefore conclude that

our approach using native protein sequences in folded nsp7-11 polyprotein better represents authentic processing reactions.

Native MS enabled characterization of the dynamic landscape of protein species, including non-covalent complexes[35]. However, this approach required using ammonium acetate as a buffer surrogate and low temperatures to prevent nsp8-mediated complex aggregation[65]. Our in-capillary experiments at a temperature of 27 °C showed decreasing reaction rates for CS10/11 over time, likely from natural substrate depletion during the ongoing process. However, influences from the elevated temperature or in-capillary acidification, which happens during prolonged nESI[37], cannot be entirely excluded. Nevertheless, linearization of data clearly reveals when the assumption of first-order kinetics is no longer valid, and $k$ can just simply be sampled from these data points. Undersampling or too little data points is fortunately not an issue in the continuous approach, as spectra were recorded at rates far higher than 10 Hz at the employed resolution of 6250. In addition, temperature-controlled experiments could yield Arrhenius plots, providing enthalpic and entropic energies of the reactions[66]. To rule out capillary-based biases, similar multi-cleavage experiments could benefit from automated liquid chromatography online buffer exchange, which would standardize the sampling timepoints and allow the reaction to be performed in standard buffer systems until the moment of sampling[67]. However, this approach would require more complex instrumentation and sample handling. Direct mass detection of intact protein intermediates enabled both the extraction of conversion rates and comprehensive insights into multi-cleavage reaction kinetics. A complementary study using cross-linking MS and hydrogen-deuterium exchange MS examined SARS-CoV-2 polyprotein processing, revealing novel spatial and dynamical information about M[pro]-nsp7-11 interactions[25]. However, these peptide-based techniques provide indirect measurements of product species. Therefore, we want to emphasize the valuable complementary information that can be gained by combining peptide-based and intact protein-based techniques in the field of structural MS.

In summary, we analyzed coronavirus polyprotein processing across multiple species, revealing dynamic intermediate products, cleavage site conversion rates, and the interconnection between processing and complex formation. Our sensitive and precise native MS approach provided novel insights into processing kinetics, demonstrating both conserved features and species-specific variations in nsp7-11 processing. We established that while complete processing enhances nsp16-nsp10 complex formation, it is not essential, and showed that functional complexes can form even between divergent hCoVs. The structural analysis of cleavage sites revealed how their structural environment contributes to processing efficiency. Our methodology demonstrates that native MS is a versatile tool for investigating simultaneous enzyme kinetics, offering advantages over conventional peptide-based approaches, enabling the detection of intact products and non-covalent complexes. This improved mechanistic understanding of coronavirus protein processing and complex formation may inform future antiviral drug development strategies targeting these essential viral processes.

## Methods

### Production of recombinant proteins

Gene sequences for nsp7-11C and nsp7-11N were taken from "Severe acute respiratory syndrome coronavirus 2 isolate Wuhan-Hu-1" as published in January 2020 (replaced by NCBI LOCUS NC_045512) and commercially synthesized (GenScript). The synthetic gene sequence for nsp7-11C/nsp7-11N with suitable overhangs were cloned with Type IIS restriction enzymes into either pASK35 + and pASK33 + (IBA life sciences), generating a plasmid with C- and N-terminal His$_6$-tag, respectively.

Gene sequences for SARS-CoV, MERS-CoV and HCoV-229E were taken from the following NCBI LOCI: R1A_SARS, NC_038294.1 and R1A_MERS1. Genes were commercially synthesized and sub-cloned via

restriction enzymes NcoI/XhoI into the vector pET-28a (+). All constructs contain a His$_6$-Strep2-SUMO-tag and are called SUMO- His$_6$-tagged for simplicity. Sequences of all constructs used are provided (Supplementary Table S7).

For expression and purification of nsp7-11 (either nsp7-11C/nsp7-11N or SUMO-His$_6$-tagged constructs), transformed BL21 Rosetta2 (Merck Millipore) were grown to OD$_{600}$ 0.4-0.6 in 2xYT medium and then induced either with 50 μM anhydrotetracycline or with 500 μM isopropyl ß-D-1-thiogalactopyranoside (IPTG) for 16 h at 20 °C. nsp7-11N and nsp7-11C and SUMO-His$_6$-tagged nsp7-11 proteins were purified via Ni-NTA affinity chromatography and Superdex10/300 (Cytiva) size exclusion chromatography (SEC)[24]. Whereby, the SUMO-His$_6$-tagged constructs were incubated with an in-house made SUMO-protease (0.1 mg protease per 1 mg target protein) and dialyzed overnight, followed by SEC.

The plasmid for nsp16 was synthesized as full-length nsp16 with N-terminal His$_6$-tag in pET22b (+) vector (Supplementary Table S1). The His$_6$-tag is followed by a short linker SAVLQ, enabling cleavage of viral protease M$^{pro}$. The plasmid for SARS-CoV-2 M$^{pro}$ in PGEX-6p-1 was generously provided by Prof. Rolf Hilgenfeld, expression and purification followed previous protocols as described in the following[68]. For nsp16 and M$^{pro}$ expression, the plasmid was transformed in BL21 (DE3). Cells were grown until an OD$_{600}$ of 0.4-0.6, cooled on ice and induced with 0.5 mM IPTG and then incubated overnight at 20 °C. Nsp16 and M$^{pro}$ were purified via Ni-NTA affinity chromatography and Superdex10/300 (Cytiva) size exclusion chromatography. To cleave the His$_6$-tag from M$^{pro}$, it was transferred into a Slide-A-Lyzer dialysis cassette (Thermo Fisher) with MWCO 10,000 and digested overnight in PreScission protease cleavage buffer (50 mM Tris HCl, 100 mM NaCl, 1 mM EDTA, 1 mM DTT, pH 8.0). PreScission protease was pulled out using GST sepharose beads to obtain pure M$^{pro}$. Nsp16 and M$^{pro}$ were flash-frozen and stored at −80 °C.

### Quality control of processing with SDS-PAGE
SDS-Page was performed with a 4–12% gradient acrylamide Bis-tris gel with XT MES running buffer (Bio-Rad Laboratories). Both constructs nsp7-11C and nsp7-11N were mixed at 36 μM with 14 μM M$^{pro}$ and incubated at 4 °C. Aliquots were withdrawn at the indicated time points and mixed with XT sample buffers to quench the reaction. Polyprotein nsp7-11 (54 μM) of the four hCoVs SARS-CoV-2, SARS-CoV, MERS-CoV and HCoV-229E were mixed with 9 μM M$^{pro}$ and incubated overnight at 4 °C. SUMO-His$_6$-tagged nsp7-11, tag-cleaved nsp7-11 with authentic termini and processed nsp7-11 of the four hCoVs were run on a 4–12% gradient acrylamide Bis-tris gel with XT MES running buffer (Bio-Rad Laboratories).

### Statistics and reproducibility
SDS-PAGE of nsp7-11 polyprotein processing from the four hCoVs was performed twice (Fig. 1B). In addition, time-resolved polyprotein processing with more time points was performed at least once for nsp7-11 of SARS-CoV-2 and nsp7-11N/C (Supplementary Fig. S2).

To assess capillary-dependent variation in native MS experiments, standard errors were derived from three technical replicates ($n = 3$). Measurements were conducted using purified proteins obtained from the same preparation. Protein samples were mixed and analyzed on separate days to account for day-to-day variability.

### Native mass spectrometry
Freshly purified samples were exchanged into a structure preserving MS compatible buffer surrogate ammonium acetate at 300 mM AmAc (99.99% purity, Sigma-Aldrich), 1 mM DTT, pH 8.0. M$^{pro}$ was exchanged into the buffer surrogate by applying two cycles of centrifugal gel filtration (Biospin mini columns, 6000 MWCO, Bio-Rad). nsp16 and nsp7-11 were exchanged by six rounds of dilution and concentration in centrifugal filter units (Amicon Ultra, 10 K MWCO, Merck Millipore).

A micropipette puller (P-1000, Sutter Instruments) was used to produce nanoESI capillaries in a two-step program from borosilicate capillaries (1.2 mm and 0.68 mm outer and inner diameter, respectively, *World Precision Instruments*) using a squared-box filament (2.5 mm × 2.5 mm). Capillaries were gold-coated by using a sputter coater (CCU-010, Safematic, $5.0 \times 10^{-2}$ mbar, 30.0 mA, 120 s, three runs to vacuum limit $3.0 \times 10^{-2}$ mbar argon). Capillaries were opened using tweezers under a microscope.

For the processing experiments, native MS was performed on a Q Exactive UHMR Orbitrap from Thermo Scientific. Positive ion mode was used by applying capillary voltages of 1.2–1.7 kV, 100–150 °C capillary temperature, 15 eV in-source CID and 25 eV in HCD cell. Trapping gas pressure optimization was set to 5 or 7. Detector optimization was set to "*low m/z*" and the ion transfer *m/z* optimization were adapted as follows: Inj. Fl. RF Ampl. to 300, Bent. Fl. RF Ampl. to 940, Trans. MP and HCD-cell RF Ampl. to 900 and C-Trap Ampl. to 2750. Tandem MS was always conducted by stepwise increase of the HCD voltage of 10–20 eV.

For the continuous approach, M$^{pro}$ was added to a final concentration of 3 μM to nsp7-11 C/nsp7-11 N (final concentration 18 μM), then the sample was briefly mixed by pipetting before transferring 1-2 μL to the capillary. Data acquisition was started 1 min after mixing. At least three replicates were conducted. The temperature of the capillary housing was 27 °C.

For the discrete approach, nsp7-11C/nsp7-11N and M$^{pro}$ were mixed with a final concentration of 20 μM and 10 μM, respectively. The mixture was then incubated on ice, and triplicate measurements were taken at selected time points. Authentic nsp7-11 of the four CoVs were mixed with the final concentration as indicated in the following Table 1.

Interaction studies of nsp7-11 of SARS-CoV-2 and MERS-CoV with authentic termini and nsp16 with cleaved and uncleaved His$_6$-tag were conducted on the Q Exactive UHMR Orbitrap. Here, fresh and frozen protein were used. For SARS-CoV-2, interaction studies were conducted with the following final concentrations: nsp7-11 with three different concentrations, 5 μM, 15 μM, 25 μM were mixed with 5 μM nsp16-His$_6$ and 3 μM M$^{pro}$ and incubated overnight. Due to the M$^{pro}$ cleavage site, nsp16 was obtained with natural termini. Unprocessed nsp7-11 (15 μM) was studied with 5 μM nsp16-His$_6$. To exclude any artefacts caused by the His$_6$-tag, nsp16-His$_6$ was also incubated with a low concentration of M$^{pro}$ (1:10). His$_6$-tag-free nsp16 was mixed with nsp7-11 at the same final concentration ratios (5 μM and 15 μM, respectively) prior to the measurements.

MERS-CoV nsp7-11 was mixed to 5 μM nsp16-His$_6$ in two different final concentrations, 5 μM, 15 μM. As described above, nsp16-His$_6$ was again incubated with M$^{pro}$ 1:10 to cleave off the His$_6$-tag and the interaction of nsp7-11 (15 μM) and nsp16 (5 μM) was tested.

Capillary temperature of the mass spectrometer was set to 100 °C, in-source CID was set to 10–20 V and HCD cell was set to 15 eV. Trapping gas pressure optimization was set to 5. Ion transfer *m/z* optimization was set to low *m/z* for the interaction experiments with processed nsp7-11 plus nsp16, and was set to high *m/z* for the interaction experiments with the unprocessed polyprotein and nsp16. Tandem MS was conducted by stepwise increase of the HCD voltage of 12.5–25 eV.

**Table 1 | Final concentrations of discontinuous processing experiments of all four CoVs**

| construct | nsp7-11 concentration | M$^{pro}$ concentration |
|---|---|---|
| SARS-CoV | 19 μM | 3.5 μM |
| SARS-CoV-2 | 19 μM | 3.5 μM |
| MERS-CoV | 19 μM | 3.1 μM |
| HCoV-229E | 17 μM | 3.1 μM |

## Data analysis

Native mass spectra were investigated, and deconvolution was supported by Unidec[69]. Deconvoluted peaks were checked, and $m/z$ ranges (gates) were noted to feed into a home-made Python script. Every gate was plotted and checked before the area under the curve (AUC) of the detected mass species was taken, which is here called intensity. The initial substrate includes five domains, resulting in five mature proteins. Thus, species intensities were normalized by using a multiplication factor corresponding to the domains or units, depending on intermediate species or mature nsps. The multiplication array was adapted depending on the species that were detected. A complete list of all detected species can be found in Supplementary Tables S1–6. As an example, it follows the multiplication array from SARS-CoV-2.

$$m = 5*I_{nsp7-11} + 4*I_{nsp7-10} + 3*I_{nsp7-9} + 2*I_{nsp7-8} + 3*I_{nsp9-11} + 2*I_{nsp9-10} + 2*I_{nsp10-11} \tag{1}$$

To correct for spray variation, the ratio of each individual species to the sum of all species was taken. For example, ratio nsp7-11 including the multiplication:

$$ratio_{nsp7-11} = \frac{5*I_{nsp7-11}}{5*I_{nsp7-11} + 4*I_{nsp7-10} + 3*I_{nsp7-9} + 2*I_{nsp7-8} + 2*I_{nsp9-10} + 3*I_{nsp9-11} + 2*I_{nsp10-11}} \tag{2}$$

Then these normalized intensities of the replicates were averaged, and the standard error of the mean calculated. The fitted rates for the CS were calculated by using the normalized intensities and summing the species containing the intact cleavage site to the corresponding cleavage site. Described below using the example of SARS-CoV-2:

Cleavage site 10/11

$$I_{10/11} = I_{nsp7-11} + I_{nsp9-11} + I_{nsp10-11} \tag{3}$$

Cleavage site 9/10

$$I_{9/10} = I_{nsp7-11} + I_{nsp7-10} + I_{nsp9-11} + I_{nsp9-10} \tag{4}$$

Cleavage site 8/9

$$I_{8/9} = I_{nsp7-11} + I_{nsp7-10} + I_{nsp7-9} \tag{5}$$

Cleavage site 7/8

$$I_{7/8} = I_{nsp7-11} + I_{nsp7-10} + I_{nsp7-9} + I_{nsp7-8} \tag{6}$$

Calculated time-dependent intensities for a given cleavage site were fitted to the first-order kinetics formula:

$$I_{fit}(t) = Ae^{-kt} \tag{7}$$

Rate constants $k$ for each cleavage were extracted from the fit.

## $K_D$ calculations

Affinities of protein-protein interactions were calculated based on the law of mass action. One binding pocket was considered for the calculation of the $K_D$s of the nsp10/16 complex, nsp10 + nsp16 ⇌ [nsp10 + nsp16]. The dissociation kinetics can be described as follows:

$$K_D = \frac{[nsp10]^*[nsp16]}{[nsp10 + nsp16]} \tag{8}$$

nsp10 or nsp16 is the concentration of the protein without a ligand. Molar fractions can be calculated by using the signal intensities of the species and the known molar concentration of nsp10 ($[nsp10]_0$) and nsp16 ($[nsp16]_0$) that were introduced to the mass spectrometer. AUC was retrieved for each peak. AUCs were assigned to the corresponding species and normalized to obtain relative signal intensities. Then, molar fractions were calculated as follows:

$$[nsp16] = [nsp16]_0 * I_{nsp16} \tag{9}$$

$$[nsp10] = [nsp10]_0 - [nsp16]_0 * I_{nsp10 + nsp16} \tag{10}$$

$$[nsp16 + nsp10] = [nsp16]_0 * I_{nsp10 + nsp16} \tag{11}$$

The Gaussian error propagation rule was used to determine the standard deviation for the $K_D$-values. $K_D$-values were calculated for each sample having different ratios. Since the replicate number was the same for each determined $K_D$, the values were simply averaged.

## AlphaFold 3 modeling and visualization

Polyprotein sequences of nsp7-11 of the four CoVs were run with standard settings from the AlphaFold3 server[70]. All models were examined with UCSF ChimeraX, and the best models were picked for comparison[71]. The best models were selected according to the overall and local confidence scores pTM and pLDDT. For the model selection, the confidence of the cleavage site areas was particularly decisive for the model selection. Thus, plotted pLDDT scores against the residues index were utilized. Furthermore, pLDDT scores were displayed by using the B-factor column of the output file. Regions with pLDDT scores higher than 70 are expected to be well predicted. Areas that show values between 50 and 70 have low confidence and should be interpreted with caution[72].

## Reporting summary

Further information on research design is available in the Nature Portfolio Reporting Summary linked to this article.

## Data availability

The MS data generated in this study have been deposited in the PRIDE database: http://proteomecentral.proteomexchange.org/cgi/GetDataset?ID=PXD049009. The corresponding extracted scans are also available in Zenodo under https://doi.org/10.5281/zenodo.15488266[73], and for an earlier version under https://doi.org/10.1101/2024.01.06.574466[74]. The uncropped gel image of Fig. 1B can be found in the source data file, and uncropped gel images of Supplementary Fig. S2 can be found at the end of the supplemental material file as Supplementary Fig. S20 and S21. Source data are provided in this paper.

## Code availability

Customized Python scripts were coded to analyze the data as described in Data analysis. Python code is available in Zenodo: https://doi.org/10.5281/zenodo.15488266[73], and https://doi.org/10.1101/2024.01.06.574466[74].

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

## Acknowledgements

We thank Alke Meents (DESY) for providing M^pro SARS-CoV-2 for preliminary MS experiments, Rolf Hilgenfeld (University of Lübeck) for providing M^pro SARS-CoV-2 plasmid for final results, Lórenza D'Alessandro for proofreading the manuscript, and Sanne Ugelstad for protein expression. We thank Susanne Witt from the protein production core facility for advice on plasmid construct design and Philipp Lewe for protein production for preliminary experiments. The Leibniz Institute for Virology (LIV) is supported by the Free and Hanseatic City of Hamburg and the Federal Ministry of Health (BMG). C.U., K.S.-K., B.K., and S.T. were funded through the EU Horizon 2020 ERC StG-2017 759661 grant. B.K. is further supported by the European Union grant 101068151, Top-AMPK, HORIZON-MSCA-2021-PF-01. F.-A.S. is supported by the Avicenna Studierendenwerk and the DFG RTG 2887 Vision. C.U. and B.K. further acknowledges funding through Bundesministerium für Bildung und Forschung (BMBF) RTK Struktur 01KI20391 and the Leibniz Association SAW-2014-HPI-4 grant. C.U., T.K., and T.D. also acknowledge funding through EU Horizon 2020 MS SPIDOC 801406 and BMBF RAC SAXFELS 05K22PSA.

## Author contributions

Conceptualization and methodology: K.S.-K., B.K. and C.U. Plasmid construction: B.K. and S.T. Protein production: K.S.-K., B.K., F.-A.S. and S.T. Provision of research materials: C.U. Investigation: K.S.-K., B.K. and S.T. Data analysis: K.S.-K., T.D. and T.K. Discussion of results: K.S.-K., B.K., C.U., T.D. and T.K. Visualization: K.S.-K. Original draft: K.S.-K., C.U. and B.K. Writing, review and editing: K.S.-K., C.U., B.K., T.D. and T.K.

## Funding

## Competing interests

The authors declare no competing interests.
