## [Transparent Peer Review file · Nature Communications]

The kinetics of nsp7-11 polyprotein processing and impact on complexation with nsp16 among human coronaviruses

Corresponding Author: Professor Charlotte Uetrecht

Version 0:

Reviewer comments:

Reviewer #1

(Remarks to the Author)

The manuscript published by Schamoni-Kast et al is focused on understanding the interplay of polyprotein processing and RTC assembly. Specifically, they studied the role of nsp11 on polyprotein processing by Mpro and the influence of polyprotein processing on core enzyme complex formation. The authors only use native MS to provide detailed and quantifiable information regarding the dynamics of CoV-2 polyprotein processing. While the manuscript is easy to read and provides detailed results and interpretations there are significant concerns with the choice of appropriate protein constructs to answer the questions they propose. The use of artificial tags, regardless of being placed on the N- or C- terminus, raises significant concerns regarding the effect on overall protein structure which is vital to determining order/rate of processing. Finally, two of the three final conclusions stated in the last paragraph of the manuscript have previously been published by others which reduces the novelty of the manuscript. Although the authors provide a new view on how nsp7-11 is being cleaved by MPro, they seem to avoid talking about prior work until the discussion, which is largely the same as what they present here. Some experiments like the SDS-PAGE they present are pretty much identical to Yadav et al Sci Adv 2022. Additionally, the authors do not address the fact that they cannot identify every cleavage product and they do not address any known conformational changes to the nsp after it has been cleaved.

Comments:

- The introduction is clear and provides good background information. While the authors present prior work on SARS-CoV processing they do not introduce the previously published work on CoV-2 processing. This would be important to place their study in the larger context of CoV-2 research.
- The authors should reframe/reword the following sentence: "It remains unclear, if the processing order of pp1a is conserved and if it is influenced by nsp11 or cs10/11" as work published by Yadav et al does address this question.
- In the first results section a conclusion states that N-terminal His-tag has no effect but no statement is made regarding C-terminal His-tag effect. This should be added to the text. Why use the His-tag, could this tag not be cleaved off prior to experiments?
- Figure 1D: Can native MS not observe nsp11? With or without the His-tag?
- Figure 2: There appears to be some discrepancies between part A and part B as part A does not appear to identify any peaks corresponding to intermediates shorter than nsp7-10, however part B plots all of the intermediates and specifically appears to show an increase in nsp7-8 which is represented in part A. Clarification of this is needed, and/or additional data is warranted in the SI as the legend states that part A this is a representative mass spectra
- How well is native MS going to be able to decipher the difference between uncleaved nsp7-8 and the nsp7-8 dimer if the masses are essentially the same and the structures are the same? Would this not influence the rate constant calculations.
- Figure 4: Does nsp16 lose the His-tag in the presence of Mpro? Part A identifies tagged nsp16 but Part B shows no tag on nsp16.
- Were the experiments with nsp16 conducted with the N-tagged polyprotein? If not, why? If yes, did this have any difference in the experiment to check for complex formation without Mpro. While it is explained that the nsp16 tag should not influence interaction the location of the C-terminal tag considering the small size of nsp11 may potentially influence interaction? Prior work by Yadav et al demonstrates that nsp11 influences the secondary structure of nsp10 thus leading to the hypothesis that the His-tag at the C-terminal may have further influence on nsp10. Additional experiments, including using alternative techniques, are needed to confirm that nsp16 and the polyprotein are needed to confidently confirm that the nsp16 does not form a complex with nsp10 while in the polyprotein.
- It is not convincing that the experiments conducted are determining whether nsp11 influences the order of processing or the

conversion of neighboring cleavage sites. As the manuscript reads it appears that in order to test this effect the authors placed a tag on either the N- or C- terminal of the protein and drew their conclusion from this. As stated, the tag improves accessibility which would suggest a change in structure and tertiary structure which plays a role in determining the rate and order of cleavage. Accordingly, this does not appear to be an appropriate way of nsp11 on processing.

- Authors state in the abstract that native MS yields sub amino acid resolution. Is this so? This warrants further explanation.

- Introduction – Last paragraph: “The dynamics of the different processing sites...”

o The dynamics have been characterized by quite extensively using HDX-MS in Yadav et al Sci. Adv. 2022

- Results; processing products and complexes – First paragraph: “sub amino acid resolution...”

o What does this mean? As in the mass differences between peaks are less than the mass of an amino acid? This doesn't make sense.

- Supplementary figure 1: This appears to be the same experiment presented in Yadav et al Sci. Adv. 2022.

- Figure 2C is relatively unclear how they arrive at their conclusions. The authors mention that they add up ‘all the detected substrates and intermediates’ to determine the reduction in cleavage site intensity, but this doesn't make much sense. These raw values should be added into the supplement with the substrates and intermediates that they use for making these calculations.

- Figure 4: No His tag on nsp16 in part B

- It seems reductive to say that there is lack of high resolution to distinguish between species. The HDX-MS experiments and the SDS-PAGE experiments (which they perform in this paper) show that there is a quite clear cleavage order. If they could detect nsp11 it would be more convinced, but it could be circumstantial that that portion gets cleaved first. In addition, the C145A mutants show that binding to cut site 9/10 happens initially. Lastly, HDX-MS detected nsp11, whereas their method could not, so it seems less convincing.

- They do nothing to address conformational changes in this work. The hypothesis proposed previously showed that there was a potential conformational change in the polyprotein, indicating that cleavage changes the protein structure. This seems to be completely ignored.

- They excluded nsp7-11N in the experiment of nsp16 formation, but they did not give a reason why they chose nsp7-11C.

- They do not have a secondary experiment to support their observations in the nsp16/10 complex formation.

-

Reviewer #2

(Remarks to the Author)

Schamoni-Kast and colleagues present an elegant study to monitor SARS-CoV-2 polyprotein processing by viral protease M-pro, using native MS. The study appears well-executed and technically sound. There is substantial overlap with an earlier study from this group (ref 12), with the main new findings focused on the determined rate constants of the polyprotein cleavage. Below are some major comments relating to the current work:

1) There is a striking difference in polyprotein processing depending on the location of the affinity tag (N- or C-terminal). This raises the question whether N-terminal tagging enhances natural rates, or C-terminal tagging inhibits it. Given the clear artefacts that the affinity tag may introduce, perhaps a construct with a cleavable tag, which would reproduce the natural termini of the polyprotein would be more appropriate.

2) On page 5, the high cleavage rate of cs10/11 is attributed to accessibility, while there are in fact differences in local sequence (LQ/A vs LQ/S, as stated in the discussion section). Also, contribution by more distant motifs in the linear sequence cannot be ruled out (also in view of the predicted structure of the polyprotein in Figure 5).

3) The temperature of the carried-out reactions is rather low, as the viral protease would normally cleave the polyprotein at physiological temperatures rather than 20 or 4 degrees Celcius. Can the authors rule out that the different cleavage sites have different temperature-dependence, which might affect the order of cleavage site processing?

4) In Figure 3C, the fit for cs10/11 is very poor (green line). Is the first-order kinetics approximation not too far from the observed rates to justify this model?

5) In the discussion section, the authors do attribute part of rate differences between cs10/11 and cs7/8 or cs9/10 to the local sequence (LQ/S vs LQ/A). This contribution could be experimentally validated by site-directed mutagenesis. This would then reveal the relative contribution of accessibility or other factors to the observed differences in rate constants.

Reviewer #3

(Remarks to the Author)

In this work, Schamoni-Kast et al. develop a native mass spectrometry workflow to study enzyme kinetics of SARS-CoV-2 main protease (Mpro), which cleaves several polypeptides in the non-structural protein domain of the virus that are necessary for viral replication. For this method, Mpro was mixed with partial length nsp (nsp7-11), and the kinetics and order of various peptide cleavage sites was determined by collecting mass spectra at various time points to compare the intensities of the full length and partial length nsp proteins. The use of native mass spectrometry is proposed as an improvement over other biochemical assays because it provides high mass resolution needed to distinguish small nsp proteins. Nsp constructs with a His tag on either C or N terminus were used to determine the effect of protein tags on enzyme kinetics. Furthermore, the authors investigate whether poly peptide cleavage is necessary for the nsp proteins to function; this assessment is based on monitoring the formation of the nsp10/nsp16 heterodimer, a non-covalent complex that can be detected by native mass spectrometry. They found that nsp10/nsp16 complex only forms if nsp10 has been cleaved by Mpro, and that the complex does not form if nsp16 is added to nsp7-11.

The study is clearly presented and well-executed. The ability to obtain time-dependent kinetic data is an admirable outcome. The work would be more impactful if the author investigated the same protein system from other coronaviruses as the

authors suggest in the discussion. These experiments seem reasonable given the high impact of the journal, and the experiments will be straightforward since the workflow is already established, and the experiments are relatively quick/easy to do, and protein expression in bacteria is simple. The Editor could comment on whether this falls beyond the scope of the paper.

On a stylistic note, the text lacks clarity in several areas with confusing language making it difficult to understand relatively simple concepts.

Specific points:

Page 2: "Since in native MS, the natural folding and protein-protein interactions are preserved¹⁸," This is an over-statement. I think the more agreed upon phrasing is that many non-covalent interactions are preserved with native MS.

Page 3: The purpose of the His tag could be more clearly explained when discussing Figure 1.

Page 3: "Hence, the N-terminal His6-tag does not impair complex formation, which confirms proper folding of the proteins." This statement is debatable. The protein could still be unfolded or partially unfolded.

Page 5: "till" should be "until 30 minutes after the start of the reaction"

Page 5: "Ultimately, the continuous processing approach works well for early kinetics, but is time limited due to heating and acidification processes within the capillary." Where and how is acidification happening? The capillary should be placed far back enough from the instrument where heating is not a problem. If the capillary is being heated, isn't this heat also influencing the early kinetic studies?

Page 6: Figure 2A: Deconvoluted spectra for nsp7-11N at different time points like nsp7-11C should be shown here or in the supporting information section.

Page 9: "This binding persisted even at a concentration approximately four times lower than the other sample." What is the "other" sample? The entire second half of this paragraph lacks clarity.

Page 9: Figure 4B: How did the His tag get removed from nsp16?

Page 11: The authors draw comparisons between their data on SARS-CoV-2 polypeptide cleavage and polypeptide cleavage by SARS-CoV data obtained by other assays. The authors should repeat their experiments with SARS-CoV to have a side-by-side comparison using a single method.

SI page 3: AlphaFold confidence score should be explained.

The charge states in the mass spectra (such as Fig 2A) should be labelled, otherwise some of the peaks are labelled with the same composition but appear at difference m/z values.

Grammatical/stylistic issues that impact clarity:

Page 1: "The polyprotein processing and complex formation is critical..." critical for what?

Page 1: "Severe-acute respiratory syndrome coronavirus 2 (SARS-CoV-2) infection can be understood as a series of dynamic molecular mechanisms that lead to multiplication of the viral genome." This generic sentence doesn't provide any new or specific information; it could be applied to any number of viruses.

Page 1: "The polyprotein internal proteases PLpro (nsp3) and Mpro (nsp5) facilitate proteolytic processing of pp1a and pp1ab into mature non-structural proteins (nsps) nsp1-nsp11 and nsp1-nsp16, respectively" This sentence is confusing because Mpro processes the (nsp5-11) part of the pp1a cleavage sites.

Page 2: "Further, the formation of the nsp7/8 complex upon processing of nsp7-8-His6 sub-polyproteins has been investigated¹³" This statement is not relevant unless the outcome of this past study is briefly summarized for readers.

Page 2: "Distinct assemblies" of what?

Page 12-13: "Assets over conventional techniques are consideration of structural context and label-free substrates that are cheap to produce." This sentence lacks clarity

Version 1:

Reviewer comments:

Reviewer #2

(Remarks to the Author)

All my comments have been addressed. The use of a construct without tags to mimic natural termini, and the inclusion of data for other coronaviruses have substantially improved the study. I have one remaining issue, which is that the authors provide their analysis code 'upon request'. I think this should rather be provided in a publicly available repository like GitHub or be included with the PRIDE deposition. The authors are advised to check editorial policy in this matter.

Reviewer #3

(Remarks to the Author)

The authors have undertaken new experiments in accordance with the reviewers' recommendations and made substantial changes to the manuscript. The new version has improved clarity and impact. Publication is recommended. I also appreciate the color-coding of the figures which makes it easier to track the different proteins/products throughout the manuscript.

Reply to reviewer comments

Reviewer #1:

The manuscript published by Schamoni-Kast et al is focused on understanding the interplay of polyprotein processing and RTC assembly. Specifically, they studied the role of nsp11 on polyprotein processing by Mpro and the influence of polyprotein processing on core enzyme complex formation. The authors only use native MS to provide detailed and quantifiable information regarding the dynamics of CoV-2 polyprotein processing. While the manuscript is easy to read and provides detailed results and interpretations there are significant concerns with the choice of appropriate protein constructs to answer the questions they propose. The use of artificial tags, regardless of being placed on the N- or C- terminus, raises significant concerns regarding the effect on overall protein structure which is vital to determining order/rate of processing. Finally, two of the three final conclusions stated in the last paragraph of the manuscript have previously been published by others which reduces the novelty of the manuscript. Although the authors provide a new view on how nsp7-11 is being cleaved by MPro, they seem to avoid talking about prior work until the discussion, which is largely the same as what they present here. Some experiments like the SDS-PAGE they present are pretty much identical to Yadav et al Sci Adv 2022. Additionally, the authors do not address the fact that they cannot identify every cleavage product and they do not address any known conformational changes to the nsp after it has been cleaved.

In response to your feedback, we have shifted our focus away from the specific impact of nsp11 to emphasize our primary contributions: (1) the determination of rate constants from untagged, folded polyprotein substrates, (2) the comparative analysis of these constants across four human coronaviruses, and (3) the structural basis (both primary sequence and higher-order structure) for the observed kinetic differences. We have revised the abstract to better highlight these key aspects of our work.

Comments:

- 1. The introduction is clear and provides good background information. While the authors present prior work on SARS-CoV processing they do not introduce the previously published work on CoV-2 processing. This would be important to place their study in the larger context of CoV-2 research.*
We extended the introduction on polyprotein processing in coronaviruses, particularly focussing on developments in recent years on SARS-CoV-2 by Yadav et al 2022 and Narwal et al. 2023 (p.3 l. 63-71).
- 2. The authors should reframe/reword the following sentence: "It remains unclear, if the processing order of pp1a is conserved and if it is influenced by nsp11 or cs10/11" as work published by Yadav et al does address this question.*
We have addressed this point by adding detailed information on p. 2-3 l.55-71. Our study's focus has evolved from specifically examining CS10/11 cleavage to conducting a comprehensive comparison of all four cleavage sites across different coronaviruses. While Yadav et al. determined cleavage order using SDS-PAGE, and their overall findings align with our observations, our native MS approach provides direct evidence of all products and intermediates with high mass accuracy. This methodology allows us to simultaneously monitor cleavage products and extract site-specific kinetic rates, offering more detailed mechanistic insights than previously available.
- 3. In the first results section a conclusion states that N-terminal His-tag has no effect but no statement is made regarding C-terminal His-tag effect. This should be added to the text. Why use the His-tag, could this tag not be cleaved off prior to experiments?*

Thank you for the comment. Based on comparing the results of our experiments with nsp7-11C- and nsp7-11N, we concluded that the His₆-tag had no effect on the preferred cleavage of the nsp10/11 site. This has now been clarified in the text p.6 l.147-151. Indeed, the N-terminally tagged construct is processed faster in the continuous approach, which could be due to adverse effects by the proximal His₆-tag in the C-terminally tagged construct.

To address the general concerns about the tag, we conducted experiments with constructs, where the purification tag was cleaved *a priori*. Expectedly, overall cleavage order in SARS-CoV-2 was not affected with cs10/11 been cleaved fastest (see new Fig. 3).

4. *Figure 1D: Can native MS not observe nsp11? With or without the His-tag?*

Native MS can observe nsp11. We included a panel in Fig. 1 with nsp11 signal showing the expected molecular weight. Optimal settings for monitoring intermediate species entailed suboptimal transmission for small peptides like nsp11, which is detected singly charged. For the calculation of the kinetic rates, it is sufficient and necessary to follow signal evolution of processing substrates and intermediates instead of mature nsps including nsp11.

5. *Figure 2: There appears to be some discrepancies between part A and part B as part A does not appear to identify any peaks corresponding to intermediates shorter than nsp7-10, however part B plots all of the intermediates and specifically appears to show an increase in nsp7-8 which is represented in part A. Clarification of this is needed, and/or additional data is warranted in the SI as the legend states that part A this is a representative mass spectra.*

Primarily, we wanted to illustrate the fast cleavage of nsp11 by showing the conversion from nsp7-11 (blue) to nsp7-10 (turquoise) in Fig. 2A. In Fig. 2B, we wanted to show the development of the species over time and included the low intensity species poorly visible in 2A (peak below *m/z* 4000, not assigned for clarity). They are however part of the kinetic rate calculations. In Fig. S3, the corresponding data for nsp7-11N is shown, due to the faster reaction, the other intermediates are of higher intensity and have been assigned accordingly.

6. *How well is native MS going to be able to decipher the difference between uncleaved nsp7-8 and the nsp7-8 dimer if the masses are essentially the same and the structures are the same? Would this not influence the rate constant calculations.*

We thank for this excellent question. Indeed, the molecular ions of uncleaved nsp7-8 and cleaved nsp7+8 heterodimer exhibit the same molecular weight. Ambiguous assignment of nsp7-8 and nsp7+8 would indeed influence the rate constant calculations, so we checked via CID whether nsp7+8 dimer was already forming. This has now been clarified in the text (p.8 l.185-190).

7. *Figure 4: Does nsp16 lose the His-tag in the presence of Mpro? Part A identifies tagged nsp16 but Part B shows no tag on nsp16.*

Yes, the His₆-tag is cleaved off upon incubation with M^{pro} because our nsp16 protein construct includes a His₆-tag with M^{pro} cleavage site. Therefore, His₆-nsp16 is converted into natural nsp16 without additional amino acids. We have included data in Fig. S10 that compares nsp7-11 complexation with both His-tagged nsp16 and untagged (precleaved) nsp16. Based on available crystal structures the His₆-tag should not interfere with complexation, we highlighted this in Fig. 5C.

8. *Were the experiments with nsp16 conducted with the N-tagged polyprotein? If not, why? If yes, did this have any difference in the experiment to check for complex formation without Mpro. While it is explained that the nsp16 tag should not influence interaction the location of the C-terminal tag considering the small size of nsp11 may potentially influence interaction? Prior work by Yadav et al demonstrates that nsp11 influences the secondary structure of nsp10 thus leading*

to the hypothesis that the His-tag at the C-terminal may have further influence on nsp10. Additional experiments, including using alternative techniques, are needed to confirm that nsp16 and the polyprotein are needed to confidently confirm that the nsp16 does not form a complex with nsp10 while in the polyprotein.

We thank the reviewer for the thorough review of the nsp16 binding experiment. Since the experiments were conducted with nsp711C, an effect on nsp10 cannot be excluded. Therefore, we performed additional experiments with untagged nsp7-11 to confirm that only cleaved SARS-CoV-2 nsp10 binds to nsp16 as shown in Fig. 5A and S10A. At higher nsp7-11 concentration ~3% complex form, however much less than for MERS-CoV-2 nsp7-11, thus it is hardly visible (Table S6). We did not use alternative techniques to further analyse nsp7-11 and nsp16 interaction as the differences between the different coronaviruses indicate that the method is capable of picking these up. Moreover, recent work by Thibert et al. (JASMS, 2024) successfully employed similar native MS approaches to study nsp16 and nsp10 interactions, further validating our methodological approach. Based on the structural protection observed in Yadav et al.'s experiments, we have added a discussion of this finding on p.11 and p.15-16 |.339-358.

9. *It is not convincing that the experiments conducted are determining whether nsp11 influences the order of processing or the conversion of neighboring cleavage sites. As the manuscript reads it appears that in order to test this effect the authors placed a tag on either the N- or C- terminal of the protein and drew their conclusion from this. As stated, the tag improves accessibility which would suggest a change in structure and tertiary structure which plays a role in determining the rate and order of cleavage. Accordingly, this does not appear to be an appropriate way of nsp11 on processing.*

The authors agree that a fusion tag could influence protein structure at the corresponding end. We considered this by comparing polyproteins with N or C terminal tags. CS7/8 was always slowest pointing to now improved accessibility in with N-terminal tag. To increase certainty, the experiments have now been repeated with a tag-free version of nsp7-11 supporting our original findings (Fig. 3 and corresponding text).

10. *Authors state in the abstract that native MS yields sub amino acid resolution. Is this so? This warrants further explanation.*

Our initial intention was to highlight the capability to resolve cleavage site positions in reaction products while simultaneously detecting natively bound cofactors (e.g., two zinc ions in nsp10) in native MS. We have removed the word “sub” as it could be misleading. Rather than specifying resolution, we now describe native MS as a “highly sensitive method allowing simultaneous monitoring of cleavage products”, which more accurately represents our analytical requirements and the technique's strengths for this application.

11. *Introduction – Last paragraph: “The dynamics of the different processing sites...”*

The dynamics have been characterized by quite extensively using HDX-MS in Yadav et al Sci. Adv. 2022

Here, we would like to differentiate between local structural dynamics and cleavage dynamics. Local protein dynamics involve small-scale movements within specific protein regions. HDX-MS characterizes these dynamics by measuring protection from deuterium uptake. Yadav et al. (Sci. Adv. 2022) applied HDX-MS to study nsp7-11 structural changes during processing, revealing varied dynamics around M^{pro} cleavage sites and distinguishing folded from unfolded regions. Their work provided valuable insights into the structural mechanisms of polyprotein-protease interactions. Protease cleavage dynamics encompass the time-dependent processes of substrate recognition, binding, and hydrolysis. Yadav et al. characterized these dynamics using SDS-PAGE

and peptide monitoring at discrete timepoints, which provided valuable insights but had limitations in mass resolution and direct product detection. Our native MS approach directly monitors all cleavage products simultaneously, offering comprehensive insight into polyprotein processing. We use the terms 'protease cleavage dynamics' and 'polyprotein processing dynamics' interchangeably, consistent with our previous peer-reviewed work. We hope this now becomes clear in the revised version.

12. *Results; processing products and complexes – First paragraph: “sub amino acid resolution...”*
What does this mean? As in the mass differences between peaks are less than the mass of an amino acid? This doesn't make sense.

This has been changed in accordance with the abstract (see 9.).

13. *Supplementary figure 1: This appears to the same experiment presented in Yadav et al Sci. Adv. 2022.*

The experiment shown in Fig. S1 (old manuscript, now Fig. S2) is methodologically similar to both Yadav et al. 2022 (Figure 1) and our own work in Krichel et al. 2020. Therefore, we have cited both previous studies (p.4 l.117-119). The SDS-PAGE serves merely as control experiment and sanity check. This is also true for the new constructs (Fig. 1).

14. *Figure 2C is relatively unclear how they arrive at their conclusions. The authors mention that they add up ‘all the detected substrates and intermediates’ to determine the reduction in cleavage site intensity, but this doesn't make much sense. These raw values should be added into the supplement with the substrates and intermediates that they use for making these calculations.*

Thank you for indicating the need for clarification at this point. We summed the relative intensities of all substrates containing the intact respective cleavage site. For example, for CS10/11, we summed the relative intensities of nsp7-11, nsp9-11 and nsp10-11 as shown in Fig. 3B. Fig. 3 in general focuses on establishing the method of quantifying and assigning cleavage substrates to a specific cleavage site to simplify to first order kinetics. The authors hope that the general restructuring of the Fig. 1 and 3 as well as the revised text make the methods for plot generation more accessible.

15. *Figure 4: No His tag on nsp16 in part B*

For a detailed answer see 9. M^{pro} cleaves off His₆-tag from the nsp16 construct.

16. *It seems reductive to say that there is lack of high resolution to distinguish between species. The HDX-MS experiments and the SDS-PAGE experiments (which they perform in this paper) show that there is a quite clear cleavage order. If they could detect nsp11 it would be more convinced, but it could be circumstantial that that portion gets cleaved first. In addition, the C145A mutants show that binding to cut site 9/10 happens initially. Lastly, HDX-MS detected nsp11, whereas their method could not, so it seems less convincing.*

Resolving power of the mass spectrometer is high enough to distinguish between species (Tamara et al., 2021, Chem. Rev). Furthermore, interaction binding studies have been repeated with Q-Exactive UHMR comprising an Orbitrap mass analyser with high resolving power. Since we improved and extended our experiments as mentioned this paragraph has changed now. Nevertheless, we apologize for the confusing wording and wish to clarify what we meant by "improv[ing] mass resolution". In native MS peak width can be reduced by simply diluting the sample, which we did here (Lu et al., 2015, JASMS).

Regarding native MS, thanks for the suggestion to rephrase and highlight the advantages of the method. Indeed, the authors think that there is more to this analysis than simply distinguish between species of cleavage products. The key strength of our approach is the detection and quantification of all products in parallel. Particularly significant is our ability to detect and

quantify nsp7-11 and nsp7-10, which we observed as the predominant species. The conversion of nsp7-11 to nsp7-10 in the beginning of the reaction is further confirmed by simultaneous nsp11 detection (see 5.). The cleavage reactions occur in parallel, and from our discontinuous experiments we can see that CS10/11 is closely followed by CS9/10. The native MS data is in this respect more precise and allows more reliable quantification than the SDS-PAGE. The authors acknowledge that the C145A mutant of M^{Pro} was an elegant way of finding CS9/10 as preferred binding site in the polyprotein protease interaction. However, it is important to note that strong binding affinity does not necessarily translate directly to early or efficient cleavage in the functional context.

Our native MS approach is complementary to HDX as employed by Yadav et al., which could detect the nsp11 peptide in isolation but not as part of various larger cleavage products and intermediates, which our method directly observes. Peptide detection inherently requires inferring the parent protein, which becomes ambiguous when multiple products or proteoforms exist simultaneously. The HDX approach was unable to detect peptides for CS8/9, and could not identify peptides for CS7/8, CS8/9, or CS10/11 after 24 hours, necessitating SDS-PAGE for confirming complete cleavage. Again, our approach hence offers complementary data, providing direct observation of intact species throughout the reaction.

17. They do nothing to address conformational changes in this work. The hypothesis proposed previously showed that there was a potential conformational change in the polyprotein, indicating that cleavage changes the protein structure. This seems to be completely ignored.

We thank the reviewer for the comment. Indeed, conformational change during polyprotein processing is an important viral strategy and of high interest. The main purpose of our work was quantification rather than conformational aspects as is now more clearly expressed in the introduction. We also properly introduce the results by Yadav et al. in the introduction (see also above) and also employ them to interpret some of our results as native MS is not reporting on the underlying structural changes upon cleavage.

18. They excluded nsp7-11N in the experiment of nsp16 formation, but they did not give a reason why they chose nsp7-11C.

The nsp16 binding experiments were now repeated with untagged nsp7-11 excluding any influence of the His₆-tag (see 10. and Fig. 5).

19. They do not have a secondary experiment to support their observations in the nsp16/10 complex formation.

Binding of nsp16 and nsp10 has been established using multiple techniques (e.g. “*Coronavirus Nsp10, a Critical Co-factor for Activation of Multiple Replicative Enzymes*” Bouvet et al, JBC, 2014). As pointed out before, the main question was if the polyprotein can also bind to nsp16. Apparently, MERS-CoV polyprotein has a higher affinity than SARS-CoV-2 polyprotein, in which mainly mature nsp10 binds, further supporting our initial results. Native MS can also determine dissociation constants K_D , which we have now added (p.11-12 l.253-260 and Fig. S12).

Reviewer #2 (Remarks to the Author):

Schamoni-Kast and colleagues present an elegant study to monitor SARS-CoV-2 polyprotein processing by viral protease M-pro, using native MS. The study appears well-executed and technically sound. There is substantial overlap with an earlier study from this group (ref 12), with the main new findings focused on the determined rate constants of the polyprotein cleavage.

We sincerely thank this Reviewer for their insightful comments and constructive suggestions, which have substantially enhanced our manuscript. Your specific suggestions regarding temperature effects and their implications for kinetic analysis improved the discussion section of our manuscript.

Below are some major comments relating to the current work:

- 1. There is a striking difference in polyprotein processing depending on the location of the affinity tag (N- or C-terminal). This raises the question whether N-terminal tagging enhances natural rates, or C-terminal tagging inhibits it. Given the clear artefacts that the affinity tag may introduce, perhaps a construct with a cleavable tag, which would reproduce the natural termini of the polyprotein would be more appropriate.*

We agree with the reviewer's assessment and have repeated the discontinuous experiments, which provides rates for all cleavage sites, with a tagless substrate making use of a cleavable tag that retained native termini. We incorporated additional early timepoints for improved temporal resolution.

- 2. On page 5, the high cleavage rate of cs10/11 is attributed to accessibility, while there are in fact differences in local sequence (LQ/A vs LQ/S, as stated in the discussion section). Also, contribution by more distant motifs in the linear sequence cannot be ruled out (also in view of the predicted structure of the polyprotein in Figure 5).*

We thank the reviewer for this important notion. We agree and present evidence that there is a broader contribution. We now include the CS sequences including flanking residues in Fig. 4 for all studied CoVs (see below). From this a clearer picture emerges, where sequence or structure could be dominating factors. It is tempting to assume that a short C-terminal extension as nsp11, would be more prone to cleavage than other more center-based cleavage sites. However, the authors recognize that especially here, in an, most likely, exposed and open cleavage motif, the primary sequence plays a pivotal role as becomes evident from distinct rate constants for CS10/11 in the other three CoVs. We have hence expanded the discussion regarding the influence of both primary sequence and structural environment on processing (starting on p.13).

- 3. The temperature of the carried-out reactions is rather low, as the viral protease would normally cleave the polyprotein at physiological temperatures rather than 20 or 4 degrees Celcius. Can the authors rule out that the different cleavage sites have different temperature-dependence, which might affect the order of cleavage site processing?*

Kinetics are supposed to be altered consistently by temperature, i.e. increased rates at elevated temperatures due to accelerated Brownian motion. However, we agree that elevated temperature could affect the stability of folds in the likely structured CS7/8, however there is no indication of such effect in the 27°C continuous data. The chosen reaction conditions at 0°C (on ice) allowed observing all cleavage sites in parallel of the naturally folded nsp7-11. Protein folding is usually preserved at such low temperatures. Our recombinantly expressed proteins, nsp7-11 and M^{pro}, have a limited shelf-life. Increasing the temperature to 37°C would result in significantly increased rate constants (~2 times relative to in capillary, ~16 times for the discontinuous assuming a rate duplication per 10°C interval) requiring usage of the continuous approach for more than 30 min with a heated sample reservoir. While the latter is easy to

implement, prolonged data acquisition in the continuous approach is not recommended due to observed unfolding and aggregation falsifying the results. While setting up the method, we observed signs of aggregation, for example fast clogging of the electrospray capillary after 1 h of cleavage at 27°C, which would be enhanced at 37°C. We have now adapted the discussion to reflect that elevated temperature could affect the cleavage order for structured CS as structural fluctuations become more frequent (p.14 l.308-310).

4. *In Figure 3C, the fit for cs10/11 is very poor (green line). Is the first-order kinetics approximation not too far from the observed rates to justify this model?*

With the applied calculation approach, the reaction is simplified to a first order kinetic.

According to the used model, the fitting works until the substrate reaches depletion, which is indeed the case in now Fig. 2C for CS10/11. Therefore, only early time points were taken into account for fitting, i.e. only the linear region was analysed before the first order kinetics break down.

5. *In the discussion section, the authors do attribute part of rate differences between cs10/11 and cs7/8 or cs9/10 to the local sequence (LQ/S vs LQ/A). This contribution could be experimentally validated by site-directed mutagenesis. This would then reveal the relative contribution of accessibility or other factors to the observed differences in rate constants.*

Thank you for the excellent suggestion. Changing amino acids could indeed be performed, but could also change folding propensity in an unpredictable manner. Rather than pursuing this approach for the current study, we implemented what we believe is an elegant alternative: comparing cleavage dynamics across four coronavirus species with naturally varied sequences. This comparative approach revealed significant differences in cleavage dynamics despite sequence similarities (Fig. 4, respective results and discussion sections). Notably, 11 of 16 cleavage sites in our substrates contain LQ/S or LQ/A motifs, yet exhibit remarkably different processing kinetics. Our observations align with previous findings (Yadav et al Sci Adv 2022) indicating that viral protease specificity extends beyond the canonical five N-terminal cleavage site residues for CS9/10. We have reserved the mutation-based approach for future investigations.

Reviewer #3 (Remarks to the Author):

In this work, Schamoni-Kast et al. develop a native mass spectrometry workflow to study enzyme kinetics of SARS-CoV-2 main protease (Mpro), which cleaves several polypeptides in the non-structural protein domain of the virus that are necessary for viral replication. For this method, Mpro was mixed with partial length nsp (nsp7-11), and the kinetics and order of various peptide cleavage sites was determined by collecting mass spectra at various time points to compare the intensities of the full length and partial length nsp proteins. The use of native mass spectrometry is proposed as an improvement over other biochemical assays because it provides high mass resolution needed to distinguish small nsp proteins. Nsp constructs with a His tag on either C or N terminus were used to determine the effect of protein tags on enzyme kinetics. Furthermore, the authors investigate whether poly peptide cleavage is necessary for the nsp proteins to function; this assessment is based on monitoring the formation of the nsp10/nsp16 heterodimer, a non-covalent complex that can be detected by native mass spectrometry. They found that nsp10/nsp16 complex only forms if nsp10 has been cleaved by Mpro, and that the complex does not form if nsp16 is added to nsp7-11.

The study is clearly presented and well-executed. The ability to obtain time-dependent kinetic data is an admirable outcome. The work would be more impactful if the author investigated the same protein system from other coronaviruses as the authors suggest in the discussion. These experiments seem reasonable given the high impact of the journal, and the experiments will be straightforward since the workflow is already established, and the experiments are relatively quick/easy to do, and protein expression in bacteria is simple. The Editor could comment on whether this falls beyond the scope of the paper.

On a stylistic note, the text lacks clarity in several areas with confusing language making it difficult to understand relatively simple concepts.

We sincerely thank the reviewer for the thorough assessment of our manuscript, identifying key limitations and suggesting valuable experiments that have substantially enhanced the quality of our work. Following the reviewer's primary recommendation as also seconded by the editor, we conducted comparative analyses across four human coronavirus species with sufficient sequence similarities to enable meaningful comparison (Fig. 4). Importantly, as suggested by the other reviewers, we performed these experiments using untagged nsp7-11 substrates to eliminate potential tag-induced artefacts. Collectively, the reviewer suggestions and our additional experiments have enabled us to refocus the manuscript on determining precise rate constants while expanding its scope to examine the structural basis, both primary sequence and higher-order structure, underlying the observed kinetic differences across coronavirus species. This also resulted in rewriting major parts of the manuscript, which hopefully now is easier to grasp.

Specific points:

1. *Page 2: "Since in native MS, the natural folding and protein-protein interactions are preserved¹⁸," This is an over-statement. I think the more agreed upon phrasing is that many non-covalent interactions are preserved with native MS.*

We agree to a certain extent and changed the wording to "Additionally, native MS preserves protein-protein interactions, enabling detection of protein complexes formed by the processing products" (p.3 l.92-93). That structure can largely be preserved has recently been shown by Esser et al.: "Cryo-EM of soft-landed β -galactosidase: Gas-phase and native structures are remarkably similar" (Sci. Adv. 2024).

2. *Page 3: The purpose of the His tag could be more clearly explained when discussing Figure 1.*

The His₆-tag was simply used for affinity purification. All three reviewers expressed concerns about how these tags impact our precise rate constant determinations. We shared these concerns, hence the initial comparison of N- and C-terminal tagged constructs. However, we now provide additional the experiments using a tag-free construct, ultimately leading to a more robust revised study.

3. *Page 3: "Hence, the N-terminal His₆-tag does not impair complex formation, which confirms proper folding of the proteins." This statement is debatable. The protein could still be unfolded or partially unfolded.*

The authors agree that the observed behaviour does not directly indicate proper folding throughout. This sentence was removed as the focus shifted away from influence of C- and N-terminal tag.

4. *Page 5: "till" should be "until 30 minutes after the start of the reaction"*
Replaced with "throughout the reaction" (p.10 l.205).
5. *Page 5: "Ultimately, the continuous processing approach works well for early kinetics, but is time limited due to heating and acidification processes within the capillary." Where and how is acidification happening? The capillary should be placed far back enough from the instrument where heating is not a problem. If the capillary is being heated, isn't this heat also influencing the early kinetic studies?*

Acidification processes are happening due to the application of high voltages in positive ion mode ("Addressing a Common Misconception: Ammonium Acetate as Neutral pH "Buffer" for Native Electrospray Mass Spectrometry" JASMS, 2017 and "Sample pH Can Drift during Native Mass Spectrometry Experiments: Results from Ratiometric Fluorescence Imaging, JASMS, 2023"). The statement with respect to heating was misleading, we rather intended to refer to the limited shelf-life of the protein at the experimental temperature. The capillary is indeed placed at sufficient distance to avoid changes in temperature. Nevertheless, we checked the temperature of the capillary housing showing 27°C. The sentence is adapted accordingly and the measured temperature is indicated (p.6 l.159-162).

6. *Page 6: Figure 2A: Deconvoluted spectra for nsp7-11N at different time points like nsp7-11C should be shown here or in the supporting information section.*

We added representative mass spectra of nsp7-11N for three time points analogous to nsp7-11C in the supporting information (Fig S3).

7. *Page 9: "This binding persisted even at a concentration approximately four times lower than the other sample." What is the "other" sample? The entire second half of this paragraph lacks clarity*
The other sample referred to the uncleaved polyprotein with nsp16. This section has now been rewritten as the experiments were improved for proper quantification and they were extended by binding experiments of MERS-CoV- nsp7-11 to SARS-CoV-2 nsp16.

8. *Page 9: Figure 4B: How did the His tag get removed from nsp16?*

The His₆-tag is lost upon incubation with M^{Pro} because our nsp16 protein construct includes a His₆-tag with M^{Pro} cleavage site. To remove potential bias, we performed parallel experiments with and without nsp16 His tag (Fig. S10) and clarified this in the text.

9. *Page 11: The authors draw comparisons between their data on SARS-CoV-2 polypeptide cleavage and polypeptide cleavage by SARS-CoV data obtained by other assays. The authors should repeat their experiments with SARS-CoV to have a side-by-side comparison using a single method.*

The authors agree with the reviewer and amongst others have included data for SARS-CoV-1 nsp7-11 using the same discontinuous approach.

10. *SI page 3: AlphaFold confidence score should be explained.*

The meaning of the confidence scores were only outlined in the manuscript and this is now replaced by a more detailed explanation (p.13 l.275-295). The confidence scores from AlphaFold3 were retrieved from the given pdb-type file. The B-factor column gives the confidence score for every atom. These confidence scores have been averaged over the reported residues.

11. *The charge states in the mass spectra (such as Fig 2A) should be labelled, otherwise some of the peaks are labelled with the same composition but appear at difference m/z values.*

Thank you for this suggestion. We included representative mass spectra of all analysed protein species with full range mass spectra assigned with name abbreviation and charge state (Fig S5 to S8). It is not possible to assign every peak as this would become too crowded but we have also revised Fig. 2A as suggested.

Grammatical/stylistic issues that impact clarity:

Page 1: "The polyprotein processing and complex formation is critical..." critical for what?

Page 1: "Severe-acute respiratory syndrome coronavirus 2 (SARS-CoV-2) infection can be understood as a series of dynamic molecular mechanisms that lead to multiplication of the viral genome." This generic sentence doesn't provide any new or specific information; it could be applied to any number of viruses.

Page 1: "The polyprotein internal proteases PLpro (nsp3) and Mpro (nsp5) facilitate proteolytic processing of pp1a and pp1ab into mature non-structural proteins (nsps) nsp1-nsp11 and nsp1-nsp16, respectively" This sentence is confusing because Mpro processes the (nsp5-11) part of the pp1a cleavage sites.

Page 2: "Further, the formation of the nsp7/8 complex upon processing of nsp7-8-His6 sub-polyproteins has been investigated¹³" This statement is not relevant unless the outcome of this past study is briefly summarized for readers.

Page 2: "Distinct assemblies" of what?

Page 12-13: "Assets over conventional techniques are consideration of structural context and label-free substrates that are cheap to produce." This sentence lacks clarity

Thanks, all these points have been addressed.